# Genomic innovations linked to infection strategies across emerging pathogenic chytrid fungi

Rhys A. Farrer[1,2,*], An Martel[3,*], Elin Verbrugghe[3], Amr Abouelleil[1], Richard Ducatelle[3], Joyce E. Longcore[4], Timothy Y. James[5], Frank Pasmans[3], Matthew C. Fisher[2] & Christina A. Cuomo[1]

To understand the evolutionary pathways that lead to emerging infections of vertebrates, here we explore the genomic innovations that allow free-living chytrid fungi to adapt to and colonize amphibian hosts. Sequencing and comparing the genomes of two pathogenic species of *Batrachochytrium* to those of close saprophytic relatives reveals that pathogenicity is associated with remarkable expansions of protease and cell wall gene families, while divergent infection strategies are linked to radiations of lineage-specific gene families. By comparing the host–pathogen response to infection for both pathogens, we illuminate the traits that underpin a strikingly different immune response within a shared host species. Our results show that, despite commonalities that promote infection, specific gene-family radiations contribute to distinct infection strategies. The breadth and evolutionary novelty of candidate virulence factors that we discover underscores the urgent need to halt the advance of pathogenic chytrids and prevent incipient loss of biodiversity.

[1] Genome Sequencing and Analysis Program, Broad Institute of MIT and Harvard, Cambridge, Massachusetts 02142, USA. [2] Department of Infectious Disease Epidemiology, School of Public Health, Imperial College London, London W2 1PG, UK. [3] Department of Pathology, Bacteriology and Avian Diseases, Faculty of Veterinary Medicine, Ghent University, Salisburylaan 133, B-9820 Merelbeke, Belgium. [4] School of Biology and Ecology, University of Maine, Orono, Maine 04469, USA. [5] Department of Ecology and Evolutionary Biology, University of Michigan, Ann Arbor, Michigan 48109, USA. * These authors contributed equally to this work. Correspondence and requests for materials should be addressed to F.P. (email: Frank.Pasmans@ugent.be) or to M.C.F. (email: matthew.fisher@imperial.ac.uk) or to C.A.C. (email: cuomo@broadinstitute.org).

Colonization of the vertebrate host niche by pathogenic fungi is an ongoing process and recent expansions of host ranges are leading to an increasing global threat to animal and human health[1]. Amphibian chytridiomycosis is emblematic of how the emergence of fungal diseases contributes to major biodiversity loss during the current sixth mass extinction[2]. Within the primitive and diverse fungal phylum of Chytridiomycota, the two causative agents of chytridiomycosis *Batrachochytrium dendrobatidis* (*Bd*) and *Batrachochytrium salamandrivorans* (*Bsal*) diverged an estimated 67 million years ago to become the only taxa known to have adapted to colonize vertebrates[3,4]. Yet, these pathogens demonstrate markedly different host species range, with *Bsal* mostly infecting a single order of hosts, the caudates (salamanders), while *Bd* infects over 700 species across all three orders of Amphibia[5,6]. These species therefore offer a unique opportunity to elucidate the acquisition of fungal pathogenicity that allowed infection of vertebrates followed by evolution of host specificity.

Host infection by both pathogens is restricted to a similar niche, the amphibian epidermis, but results in markedly different outcomes (focal necrosis and ulceration in *Bsal* versus hyperplasia and hyperkeratosis in *Bd*)[4]. We therefore predict that the jump of Chytridiomycota to invade amphibian hosts has been facilitated by the acquisition of common ancestral traits, whereas subsequent differentiation of infection strategies has been the result of lineage-specific adaptations. To assess these predictions, we here report the sequencing of the genomes of *Bd* and *Bsal* and comparison to those of two related saprobic chytrids; *Homolaphlyctis polyrhiza* (*Hp*) is in the same order (Rhizophydiales) as *Batrachochytrium* and is the closest, so far known, relative to *Bd*, while *Spizellomyces punctatus* (*Sp*) is in a different, primarily terrestrial, order (Spizellomycetales). Differences in protease gene content for the two pathogens are then measured experimentally, revealing stage specific activity. To more broadly characterize the host–pathogen interplay during infection, we compare the *in vivo* transcriptomes of *Bd* and *Bsal* in a susceptible model host species (*Tylototriton wenxianensis, Tw*) at late stage of infection against transcription *in vitro*; this reveals fundamental differences between these pathogens related to their differing infection strategies and the attendant host response.

## Results

**Genome content of *Batrachochytrium* species.** We find the evolutionary adaptation to infect amphibians to be correlated with the acquisition of genes, encoding secreted proteins that are unique either to the genus *Batrachochytrium,* or to *Bd* or *Bsal*. By sequencing both *Bd* and *Bsal* (Supplementary Table 1), we find that genome size of the host-restricted *Bsal* is larger (32.6 Mb) and more complex than that of the broad host range *Bd* (23.7 Mb) as well as those of *Hp* (16.7 Mb) and *Sp* (24.1 Mb), two-related free-living chytrid saprobes (Fig. 1). This variation in genome size is reflected in the complexity of the *Bsal* genome, with 10,138 protein-coding genes predicted compared to a range of 6,254–8,952 for the other chytrids (Supplementary Table 2, Supplementary Fig. 1c,d). *Bd* and *Bsal* share a core set of 5,706 gene clusters (6,403 or 6,344 genes, respectively, including paralogs), of which 542 clusters are not found in the two saprobic chytrids and include specific functions related to cell wall modification and candidate secreted effectors (discussed below) (Supplementary Fig. 1c, Supplementary Tables 3 and 4). The two *Batrachochytrium* species are most closely related and share large regions of synteny; both species are more closely related to *Hp* than to *Sp*, and *Batrachochytrium* and *Hp* share a larger proportion of syntenic orthologs than either species shares with *Sp* (Fig. 1).

Similar to *Bd*, heterozygous positions were abundant in *Bsal* and found throughout the genome (Supplementary Fig. 2a, Methods). This level of heterozygosity (161,831 total; 4.96 kb) is similar to those found in the non-GPL *Bd* isolates using the same methods (for example, *Bd*CH ACON 141,620 = 6.05 kb, *Bd*CAPE TF5a1 122,352 = 5.23 kb), while *Bd*GPL strains have less heterozygosity (for example, *Bd*GPL JEL423 50,711 = 2.16 kb)[7]. Similar to *Bd*, the *Bsal* genome appears to be diploid, with potential examples of trisomy, based on depth of coverage, allele balance and SNP phasing (Methods).

**Protease family expansion and activity.** Several of the expanded and lineage-specific protein families are highly expressed during *in vivo* infection of the salamander *T. wenxianensis* (*Tw*). Using RNA-Seq, we compared gene expression for parallel infections by *Bsal* and *Bd* or growth of each pathogen in culture. Of the chytrid genes that were significantly upregulated *in vivo* (n = 550), a large proportion was unique to *Bsal* (n = 327; 60%), unique to *Bd* (n = 43; 8%) or unique to the genus *Batrachochytrium* (n = 44; 8%). Furthermore, about half of the *Batrachochytrium*, *Bsal* and *Bd* upregulated genes were secreted (55% and 47%, respectively). The fact that these secreted proteins are both largely not present in the saprobic chytrids based on ortholog identification, and that they show increased transcription during host colonization, suggests that the transcriptional response is focused on a unique host-interaction strategy in each species. Further, these upregulated genes likely include key virulence factors, acquired for colonization of a specific host species group for each pathogen within a vertebrate host class. Indeed, these include the M36 metalloproteases implicated in pathogenicity in chytrids[7,8] as well as at least two large families of secreted proteins (Tribes 1 and 4) with no recognizable functional domains, which are very highly expressed and may represent novel virulence factors unique to *Bsal* (Fig. 1, Supplementary Figs 3 and 4).

The M36 metalloprotease family known to mediate host invasion in other systems[9,10] is markedly expanded in *Bsal* (Supplementary Fig. 5), concordant with the aggressive necrotic pathology that this pathogen causes. Both *Bsal* (n = 110) and *Bd* (n = 35) have expanded M36 families compared to lower counts in the free-living saprobic chytrids *Sp* and *Hp* (n = 2 and 3, respectively). Genus and species-specific expansions in the M36 metalloproteinase family suggest both ancestral gene-family expansions underpinning adaptation to vertebrates, and species-specific expansions that are potentially contributing to the delineation of the host species group and pathogenesis. Phylogenetic analysis revealed a subclass of closely related M36 metalloproteases that are shared across both pathogens that we term the *Batra* Group 1 M36s (G1M36) (Fig. 2a). Species-specific gene-family expansion is illustrated by the presence of a novel secreted clade of M36 genes (n = 57) unique to *Bsal*, which we have termed the *Bsal* Group 2 M36s (G2M36) (Supplementary Table 5). These G2M36s are entirely encoded by non-syntenic regions of the *Bsal* genome (Fig. 1), supporting a recent species-specific expansion. Although most G1M36s and G2M36s are strongly upregulated in salamander skin, eight *Bsal* G1M36s (19%) appear more highly expressed *in vitro* (Fig. 2b), suggesting complex regulatory circuits underlie this subclass of protease in *Bsal*. Regulation and activity of the expanded proteases is complex and life stage specific (Fig. 2c,d, Supplementary Table 6). The upregulation of secreted G1M36s and the associated protease activity in *Bd* zoospores compared to *Bd* sporangia points to a crucial role of these proteases during early host colonization in *Bd*, for example, during insertion of their germ tube into the epidermal cells[11]. In contrast, the low

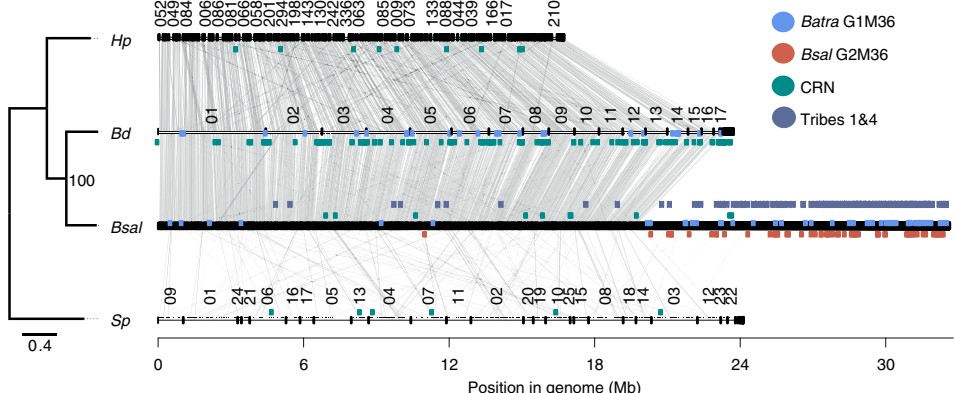

**Figure 1 | Relationship and genomic organization of four chytrid species.** A phylogenetic tree inferred using RAxML indicates the relationships of the four chytrids (branch lengths indicate the mean number of nucleotide substitutions per site). To the right is a synteny plot, visualizing regions that span two or more orthologs between any two species as a connected grey line. Scaffold numbers are shown above each genome axis if longer than 100 kb, and the location of *Batra.* Group 1 M36s (G1M36), *Bsal* Group 2 M36s (G2M36), Crinklers (CRN) and the secreted upregulated Tribes 1 & 4 are depicted with coloured bars.

protease activity in *Bsal* zoospores, but high activity in the maturing *Bsal* sporangia suggests a role during later stages of pathogenesis, for example, in breaching the sporangial wall of developing sporangia and subsequent spread to neighbouring host cells[4].

**Variation of cell-surface proteins in *Bd* and *Bsal*.** Another notable family of proteins markedly expanded in both *Bd*[12] and *Bsal* compared to the free-living chytrids is characterized by multiple copies of the CBM18 domain (Fig. 3a). This domain is predicted to bind chitin and most copies of these proteins contain secretion signals that will target them to the cell surface or extracellular space. Species-specific differences are notable in the pronounced truncation of the lectin-like CBM18s of *Bsal*, suggesting a fundamental difference in capacity to bind some chitin-like molecules. In comparison, CBM18 genes in *Bd* are three-fold longer and harbour on average eight CBM18 domains compared with only 2.6 for *Bsal*. The lack of any significant changes in regulation of CBM18 upon exposure of sporangia to chitinases renders their role in protecting the fungi from host chitinase activity by fencing off the fungal chitin unlikely (Fig. 3b, Supplementary Table 6). Rather, we hypothesize that the CBM18s play a role in fungal adhesion to the host skin or in dampening the chitin-recognition host response.

**Expanded content of repetitive elements.** Expansions of the M36 metalloprotease and CBM18 gene-family expansions coincide with an increased occurrence of repeat-rich regions in both pathogens. The fraction of the genome classified as repetitive sequence is 17% and 16% for *Bd* and *Bsal* compared to 3.7% and 4.5% for *Sp* and *Hp*, respectively (Supplementary Fig. 2). However, the dramatic differences in composition of repetitive regions between *Bd* and *Bsal* suggest independent acquisition of repeat-rich regions by both pathogens. While *Bsal* is rich in Alu elements (1.8 Mb; 5.6% of the genome), these are completely absent in *Bd*, *Hp* and *Sp*. Conversely, *Bd* is rich in DNA elements and long tandem repeats (LTR) (2.5 Mb; 10.9% of the genome), which are massively reduced or even absent in the other three chytrids. These differences suggest independent acquisition and diversification of repeat-rich regions contributing to genomic diversification of these species. The lack of the RNA-dependent RNA polymerase involved in RNAi defence may contribute to

proliferation of repetitive elements in *Bd* and *Bsal*, though this gene is also missing from *Hp*.

The known association of gene-sparse, repeat-rich regions of the genome with high densities of virulence effectors in well-studied eukaryotic plant pathogens[13] led us to examine gene-sparse regions of *Bd* and *Bsal*. Notably we found regions of low-gene density include homologues of a class of virulence effectors termed Crinkler and Necrosis (CRN) genes, previously found in the *Phytophthora* and *Lagenidium* genera of Oomycetes[14,15]. CRN-like genes in *Bd* had long intergenic regions consistent with a gene-poor repeat-rich environment (averaging 1.4 kb; Supplementary Fig. 6, Supplementary Table 7)—a trait shared with *Phytophthora infestans* T30-4 (ref. 16). While previously noted in *Bd*[7,17], we find the CRN-like family is more widely distributed among the Chytridiomycota than previously realised. We identified 162 CRN-like genes in *Bd*, 10 in *Bsal*, 11 in *Hp* and 6 in *Sp* (Fig. 4a), many of which ($n = 55$) belong to a single subfamily (known as DXX); genes in *Bd* contain one of two N-terminal motifs (Supplementary Fig. 7). As CRN-like genes did not appear highly expressed during advanced *Tw* infection (Fig. 4b), we next examined expression during additional life cycle stages. In both *Bd* and *Bsal*, some CRN-like genes were more highly expressed in the zoospore life stage compared to the sporangia life stage (Fig. 4c, Supplementary Table 6). However, incubation of *Bd* zoospores with *Tw* tissue for 2 h showed an increased expression of CRN genes, whereas *Bsal* zoospores were associated with decreased expression, indicating that CRN genes are of interest in the early infection stage of *Bd*, but not *Bsal*; the notable expansion of CRN-like genes in *Bd* suggests that they are of importance, however their function remains unknown. These data corroborate recent findings showing that the CRN-like family occur broadly though patchily throughout microbial eukaryotes as effectors in inter-organismal conflicts[18].

**Differences in host immune response.** Despite *Bsal* and *Bd* being their own closest known relatives and causing a similar lethal skin disease, both vertebrate chytrids deploy strikingly different strategies during pathogenesis, reflecting their individual infection strategies in salamander hosts. The relatively mild skin pathology caused by *Bd*[19] nevertheless induces a massive host response resulting in epidermal hyperplasia and hyperkeratosis coinciding with marked expression changes in genes involved in epidermal cornification, electrolyte and fluid homeostasis and immunity (Fig. 5). To examine host gene expression, we generated a *de novo*

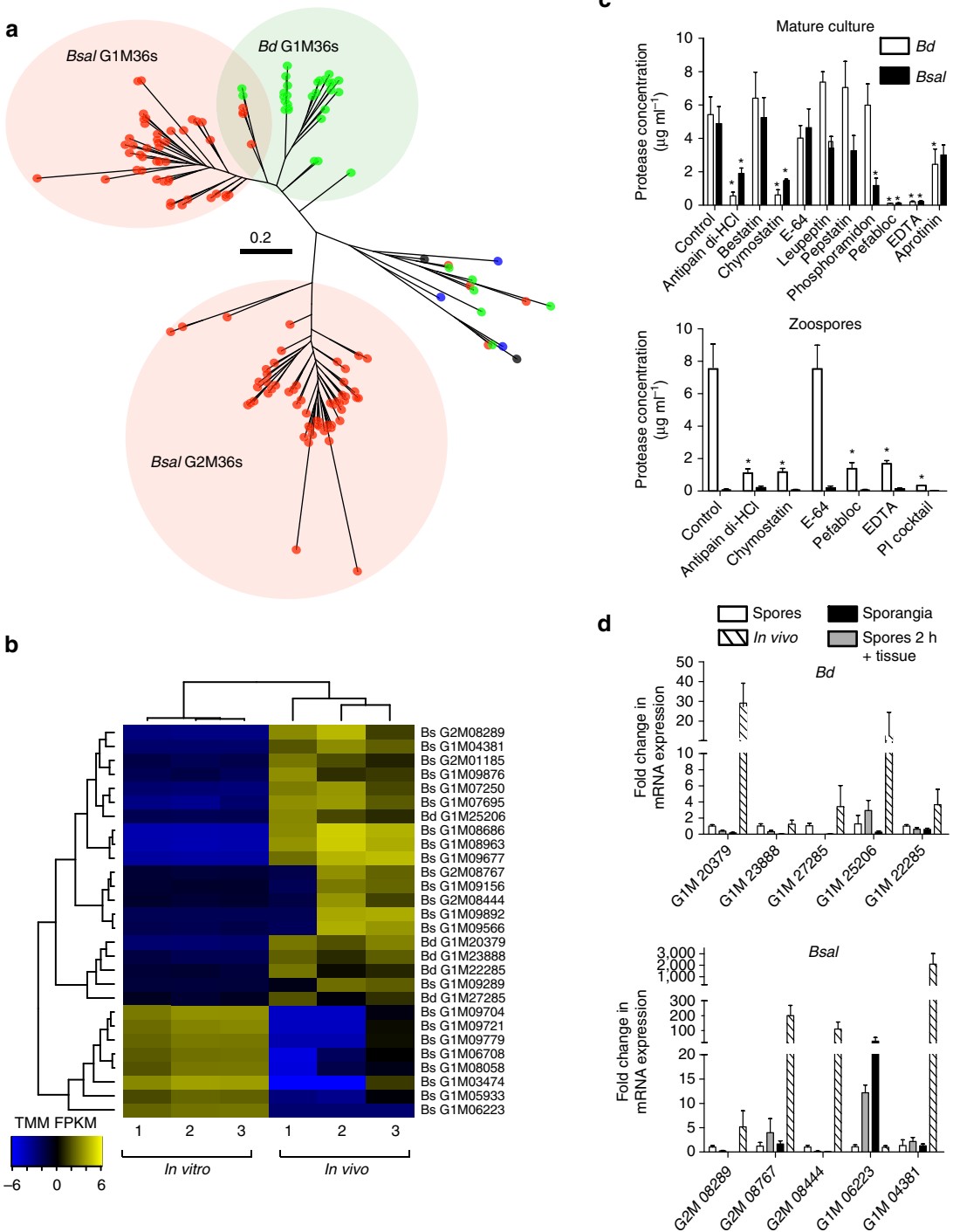

**Figure 2 | M36 gene-family relationship and expression at differing life stages.** (**a**) A phylogenetic tree inferred using RAxML from protein alignments of all identified M36 proteins in the four chytrids (branch lengths indicate the mean number of nucleotide substitutions per site). *Bd* genes are shown in green, *Bsal* genes in red, *Sp* genes in blue and *Hp* genes in black. (**b**) M36 expression was calculated (TMM normalized fragments per kilobase mapped; FPKM) across three *in vitro* replicates and three *in vivo* replicates (shown as 1, 2 and 3). Only M36 transcripts that are significantly differentially expressed in *Tw* are shown. A greater number are significantly differentially expressed in *Bsal* compared with *Bd*, which also include eight *Bsal* G1M36s that are downregulated. Trees indicate hierarchical clustering between data sets (above) and genes (left of heatmap). (**c**) Protease activity in *Bd* and *Bsal*. The protease concentration was determined in mature cultures (above) and spores (below) that were incubated with unsupplemented distilled water (control) or distilled water supplemented with different protease inhibitors. The results are presented as means + s.e.m. using a non-parametric Mann–Whitney *U*-analysis. Significant changes compared to the control group are signed with an asterisk (of $P < 0.05$). (**d**) G1M36 and G2M36 mean fold changes in mRNA expression profiles in *Bd* (above) and *Bsal* (below). The data shows the normalized target gene quantities in spores that were incubated with skin tissue of *Tw* for 2 h, 3-day-old sporangia grown in TGhL and skin tissue from chytrid-infected *Tw* animals relative to freshly collected spores. The results are presented as means + s.d. Significant differences in expression between each experimental group are shown in Supplementary Table 6.

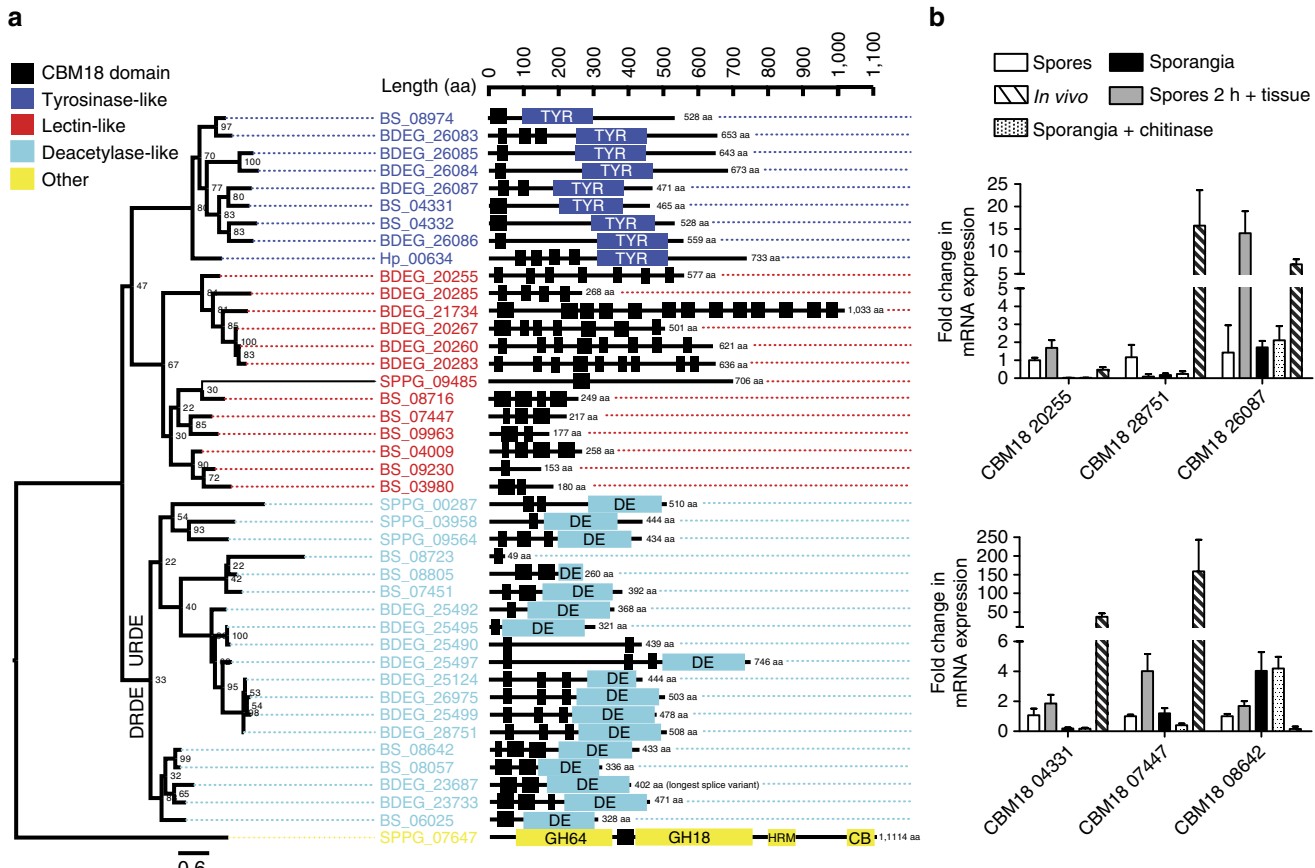

**Figure 3 | CBM18 sequence similarity and expression at different life stages.** (**a**) A phylogenetic tree of all chytrid CBM18 proteins inferred using RAxML with the per cent of 1,000 bootstrap replicates indicated for each node (branch lengths indicate the mean number of nucleotide substitutions per site). To the right of gene names are a diagram of the domain structure (black boxes indicate CBM18 domain, colours indicate larger domains shown in legend). (**b**) CBM18 mean fold changes in mRNA expression profile in *Bd* (above) and *Bsal* (below). The data represent the normalized target gene amount in spores that were incubated with skin tissue of *Tylototriton wenxianensis* (*Tw*) for 2 h, 3-day-old sporangia grown in TGhL, 3-day-old sporangia treated with TGhL broth supplemented with chitinase for 2 h and skin tissue from chytrid-infected *Tw* animals relative to freshly collected spores which is considered 1. The results are presented as means + s.d. Significant differences in expression between each experimental group are shown in Supplementary Table 6. DRDE, down-regulated deacetylase-like; UPDE, up-regulated deacetylase-like.

assembly of *Tw* transcripts based on RNA-Seq data (Supplementary Table 8). *Bd* infection induces marked upregulation of *Tw* host genes involved in innate (that is, inflammatory, antimicrobial peptides) and adaptive (that is, immunoglobulin, MHC) immune responses, a feature of infection that has previously been noted in some species for *Bd*[20]. Simultaneously, mucins are downregulated, and an absence of infiltration by immune cells in the infected skin was found. These responses suggest the host immune defences are dysregulated, which combined with loss of homeostasis, explains the lethality of the infection. In contrast, salamanders remain basically unresponsive towards the *Bsal*-induced necrosis and massive tissue destruction[4] that results in erosion of the skin barrier leading to subsequent, overwhelming septicaemia and death. Unlike the significant upregulation of immune genes in response to *Bd* infection, we detected very little increased transcription linked to an immune response to *Bsal* (Fig. 5c). The lack of a substantial host immune response to *Bsal* suggests that it has immune-dampening properties in caudates, which could be attributed to one of its unique gene families.

## Discussion

Our study shows that the differential expansion of putative virulence factors by at least two chytrid fungi is associated with an expanded host range of chytridiomycete fungi to include vertebrate hosts. Despite being closely related and both causing a lethal skin disease, the pathogenesis of these vertebrate-adapted fungal pathogens is strikingly different with the more host-restricted pathogen deploying a notably expanded arsenal of putative virulence factors and yet not triggering a strong host immune response. The evolutionary plasticity that has resulted in this remarkable, and thus-far uncharacterised, arsenal of fungal virulence factors within the genus *Batrachochytrium* underscores the urgent need to uncover and describe the global phylogenomic diversity of these fungi.

Pressing challenges include an understanding of the population genetic structure of *Bsal*, and variation in virulence that may exist. In this experiment, we have used a widely used isolate of *Bd* (JEL423) belonging to the Global Panzootic Lineage (*Bd*GPL), which is responsible for major amphibian die-offs[21]. However, other *Bd* lineages and other hosts may yield variation in induced immune responses alongside life stage or infection-specific transcriptional responses. Furthermore, the experimental validation of many of the putative pathogenicity factors in these lineages and species of chytrids remain to be demonstrated. In particular, the newly discovered tribes in both *Bd* and *Bsal*, as well as the enigmatic chytrid Crinkler genes, currently have no ascribed function and further research on these putative virulence effectors is urgently needed. Finally, the difficulty we

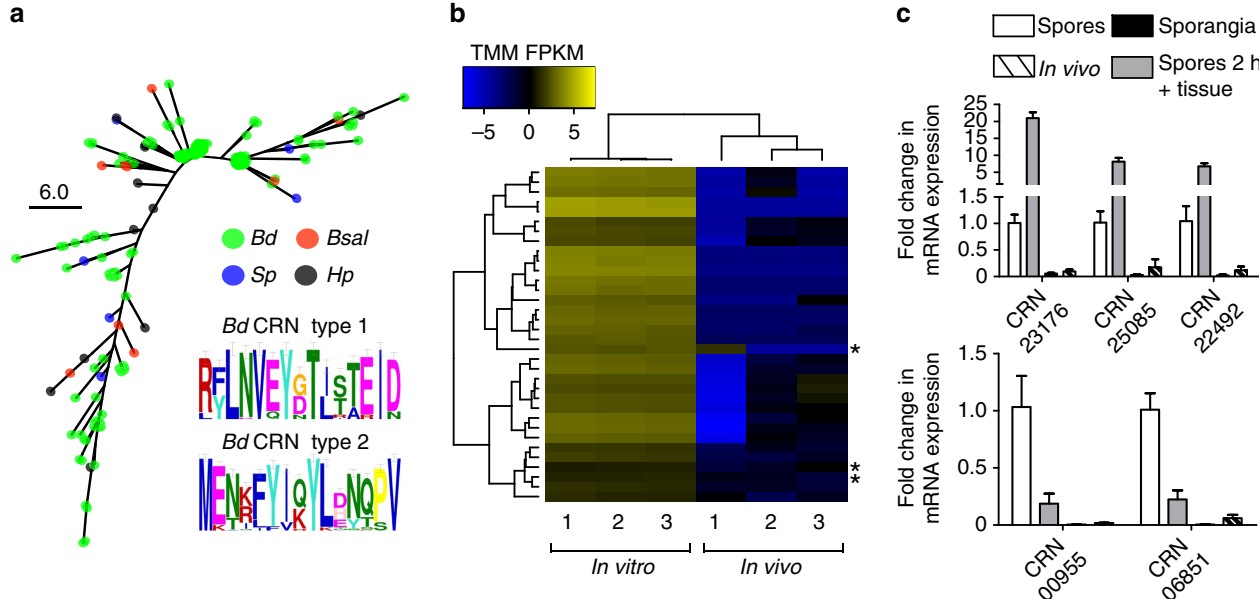

**Figure 4 | Crinkler gene-family relationship and expression patterns. (a)** A phylogenetic tree inferred using RAxML from the N-terminal region (first 50 amino acids) of all identified CRNs, showing *Bd* genes in green, *Bsal* genes in red, *Sp* genes in blue and *Hp* genes in black (branch lengths indicate the mean number of nucleotide substitutions per site). Two motifs were identified from these sequences. **(b)** Most CRNs were downregulated in *Tw*, and all those that were significantly differentially expressed (shown) were downregulated, including three *Bsal* genes (indicated by asterisk). Trees indicate hierarchical clustering between data sets (above) and genes (left of heatmap). **(c)** CRN mean fold changes in mRNA expression profile in *Bd* (above) and *Bsal* (below). The data show the normalized target gene amount in spores that were incubated with skin tissue of *Tw* for 2 h, 3-day-old sporangia grown in TGhL and skin tissue from chytrid-infected *Tw* animals relative to freshly collected spores which is considered 1. Significant differences in expression between each experimental group are shown in Supplementary Table 6.

faced in generating a contiguous *Bsal* assembly could be improved through longer sequencing reads, or techniques that were used to ensure the high quality *Bd* genome. Indeed, an improved *Bsal* assembly may further resolve repeat families, syntenic divergence from *Bd*, and aneuploidies that may be present, which in turn may affect expression, life-stage and infection dynamics.

Understanding the amphibian-destroying armamentarium that chytrids harbour is crucial to making informed predictions of the risk that novel chytrid lineages pose to naïve and thus-far uninfected regions of the world. This research is a vital step towards correctly informing policy makers of the risk that specific lineages pose, enabling legislation to be enacted to stop the further advance of these pathogens into disease-free regions through biosecurity mechanisms such as restricting animal trade[22].

## Methods

**Genome sequencing and assembly.** *B. salamandrivorans* (*Bsal*) was sequenced using 29,503,468 paired-end reads (101 nt long) with insert sizes of 441 bp, and an additional 5,895,159 reads with insert sizes ranging from 500 bp to 2.5 kb, with a mean of 900 bp (totalling 262.9 × depth; Supplementary Table 1). The genome was assembled with Allpaths version R48559 (ref. 23) using 140 × fragment coverage. We detected high levels of heterozygosity, so included the 'haploidify' setting. The completeness, contiguity and correctness of the assembly were analysed using the GAEMR genome analysis package (http://www.broadinstitute.org/software/gaemr/). We also assessed the ability of Platanus v1.2.1 (ref. 24), which is designed for the assembly of highly heterozygous genomes. However, this tool did not surpass the Allpaths assembly. Additionally, we attempted a number of kmer-normalized assemblies, and splitting MiSeq reads to make pseudo-jump-paired reads. In general, these assemblies had fewer contigs but also less total sequence (dropping 2–5 Mb of sequence). We therefore chose to use the Allpaths assembly that was 32,636,440 nt in length, divided among 5,358 contigs (*N50* 10.5 kb).

*B. dendrobatidis* (*Bd*) strain JEL423 was sequenced using Sanger technology and paired-end reads were generated from 4 kb plasmid, 10 kb plasmid and 40 kb Fosmid genomic clones. A total of 12.5 × sequence coverage was generated and assembled using Arachne[25]. A 175 kb scaffold corresponding to the mitochondrial genome was removed from this assembly for annotation and analysis. Nearly all of the sequence is in large scaffolds; 98% of the sequence is present in 17 scaffolds; 97% of the bases of Q40 quality are higher (one error every $10^{-4}$ bases).

**Heterozygosity and ploidy of *Bsal* in relation to other chytrids.** Genome-sequencing reads from *Bsal* were aligned back to the *Bsal* genome using Burrows Wheeler Aligner (BWA) v0.7.4-r385 mem[26], providing an average 169 × deep alignment. Binomial-SNP Caller from Pileup format (BiSCaP) v0.11 (ref. 27) with default settings was used to call variants. *Bd* strains harbouring trisomic and tetrasomic chromosomes had a high number of bi-allelic heterozygous positions with tri-allelic binomial probabilities (33:66 ratios) (between 1,944 and 6,654 per isolate[7]) and very few tri-alleles (between 2 and 145 per isolate[7]) possibly suggesting transient chromosome loss and gain as seen by in vitro passages of *Bd*. Even more pronounced were the 65,642 heterozygous positions in 33:66 ratios (40.56% of all heterozygous calls) across the *Bsal* genome. However, only 38 bases had three separate alleles, 24 of which included insertion/deletion variants.

The majority of *Bsal* heterozygous positions (136,501; 84%) were phased into two distinct haplotypes of two or more positions, all of which had 100% of the reads in accordance (using a minimal depth of 4 for each haplotype). These values are consistent with a polyploid genome arising from recent or transient duplications of one (or both) sets of the homologous chromosomes. Ploidy of the *Bsal* genome was further analysed using depth of coverage and allele balance (tallies of bases with differing per cent of reads agreeing with the reference: 47–53% for bi- or tetra-alleles, 30–36% and 63–69% for tri-alleles, and 22–28% and 72–78% for tetra-alleles). Due to large number of contigs, they were ordered according to mean depth of coverage. Contigs with the lowest and highest mean coverage were over represented by the smallest contigs. All contigs smaller than 5 kb were therefore excluded from representation (Supplementary Fig. 2b).

Approximately 1/8th of the genome with the lowest coverage had the greatest proportion of tri-allelic positions. The remaining 7/8th of the genome had predominantly bi-allelic heterozygous positions, consistent with a tetraploidy. This pattern of tetraploidy with trisomy is similar to those identified in *Bd* JEL423 (ref. 7). However, due to the low contiguous nature of the assembly, the 1/8th assembly that has evidence for triploidy could be enriched for assembly issues, which may suggest an overall diploid genome. Indeed, mapping synteny and ploidy together did not suggest whole chromosome copy number variation, and instead that tri-alleles may derive from repetitive elements (such as centromeres), or otherwise problematic regions of the assembly (Supplementary Fig. 2b). To examine this, we performed two-tailed Fisher's exact test with *q*-value FDR[28]

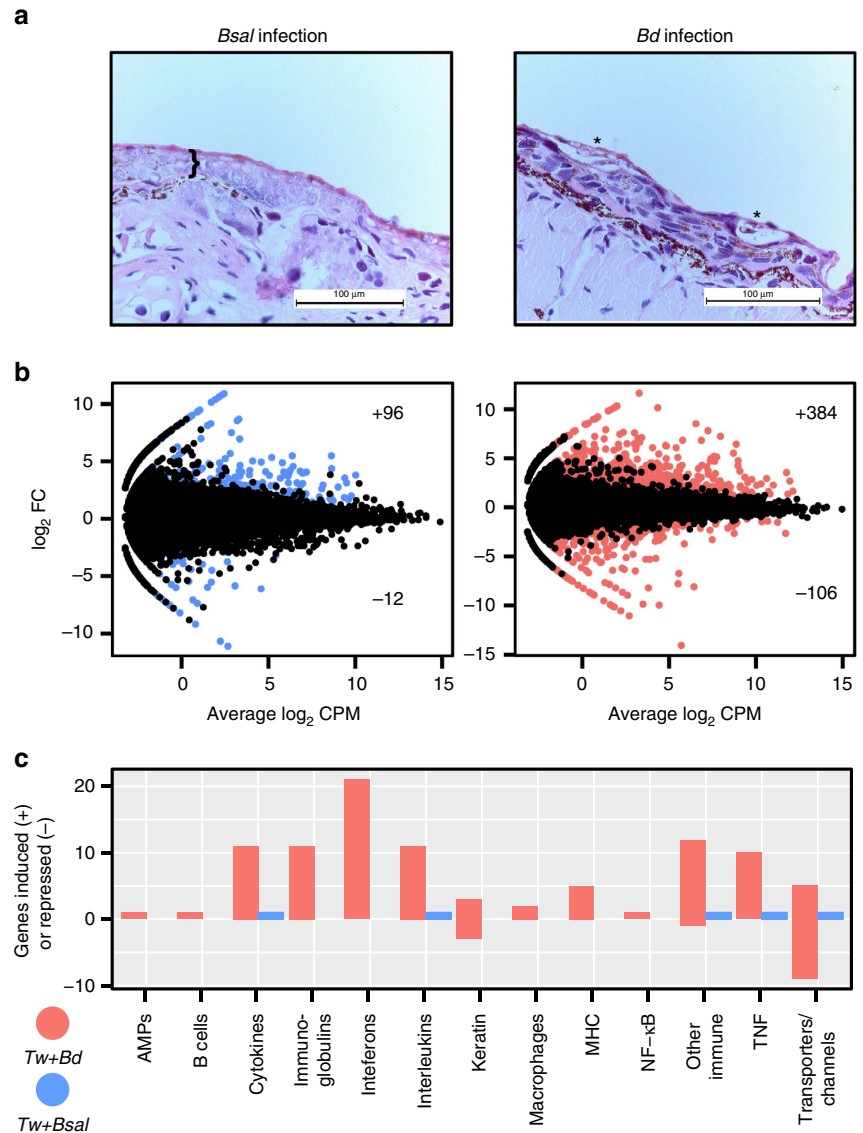

**Figure 5 | Transcriptomes and skin histology for *Tw* post infection from *Bsal* or *Bd*.** (**a**) Skin of *Tw* at 10 days after exposure to *Bsal* (left) or *Bd* (right). Bsal thalli are abundantly present across the entire thickness of the epidermis (}) resulting in extensive loss of keratinocytes, whereas *Bd* thalli are associated with the superficial epidermal layers and hyperkeratotic foci (*). For both infections, histological evidence of an inflammatory response is lacking (Hematoxylin and eosin stain, ×400). (**b**) MA plots (showing Log$_2$ fold change in the trimmed mean of *M*-values (TMM) normalized Fragments Per Kilobase of transcript per Million mapped reads (FPKM) vs average Log$_2$ counts per million) of *Tw* transcripts during *Bsal* infection (left) and *Bd* infection (right) compared with non-infected. Significant differential expression is highlighted in blue (*Bsal*) and red (*Bd*), where FDR *P* value < 0.001 and > fourfold change of TMM normalized FPKM. (**c**) Multiple classes of immune genes (*x*-axis) were found differentially expressed during *Bd* infection, while few were found during *Bsal* infection. The *y*-axis shows the number of genes either upregulated (positive count) or downregulated during infection (negative count).

of PFAM domains contained in tri-allelic compared with bi-allelic regions. We identified 13 significantly enriched terms, four of which were Glutamine amidotransferases (GATases 2, 4, 6, 7). Other likely multi-copy family members included HSP90, ABC transporters and a peptidase S41.

**Repetitive content of *Bsal* in relation to other chytrids.** To characterize the repetitive content of each of the four chytrids, we ran RepeatModeler v1.0.7 (http://www.repeatmasker.org) to identify *de novo* repeats from the assemblies. The proportion of repetitive elements between the two *Batrachochytrium* species was relatively consistent (*Bd* = 17.1%, *Bsal* = 16.2%). However, the total repetitive sequence in *Bsal* amounted to an additional 1.3 Mb, and both *Batrachochytrium* genomes were repeat rich compared with *Sp* (3.7%) or *Hp* (4.5%). Furthermore, the composition was markedly different between the two species (Supplementary Fig. 2c,d). Specially, *Bsal* has about 1% (263 kb) of its genome comprised of short interspersed nuclear elements, all of which were Alu elements, which is entirely absent in *Bd* and *Hp*, and comprises < 0.01% of the *Sp* genome. *Bsal* also had > 10 × the number of low complexity repeats (192 kb:13 kb) and nearly 20 × the genomic occupancy of simple repeats compared with

*Bd* (404 kb:36 kb). Conversely, *Bd* had 2 × the occupancy of long interspersed nuclear elements 2 in its genome compared with that found in *Bsal* (195 kb:122 kb). Even more pronounced was the difference in LTR (*Bd* = 411 kb, *Bsal* = 153 kb) and DNA elements (*Bd* = 1.3 Mb, *Bsal* = 0.5 Mb). DNA elements were not identified in either of the non-amphibian-infecting chytrids. That *Batrachochytrium* should harbour such repeat-rich genomes compared with its closest chytrid relatives may indicate a conserved functional role, such as gene duplication, or an ancestral reduced ability to purge repetitive elements.

The largest difference in repetitive content was in the number of unclassified repeats identified in *Bsal* (4.2 Mb; 13%) compared with *Bd* (2.2 Mb; 9.24%). To characterize these repeats, we first ran an all vs all BLASTn with e − 50 to avoid spurious hits. OrthoMCL and manually checking both confirmed a number of unknown families that could be merged with either other unknown families or annotated families. One repeat family in *Bd* had sequence similarity with LTR Copia at the start, and DNA elements on the other-end (unknown families were therefore placed into an LTR/DNA category). We next took all remaining unknown families that comprised > 0.3% of each genome, identified ORFs using ORFFinder, and BLASTp the largest ORF to the nr protein database. A number

of unknown families were categorized including *Sp* repeat-family 6, which contained five domains belonging to LTR copia-type transposons. Similarly, *Sp* family 71 had four domains belonging to Gypsy transposons. This process gave categories for 92% of *Sp* repeats, 90% *Hp* repeats, 75% of *Bd* repeats and 62% of *Bsal* repeats (Supplementary Fig. 2c). Those that were not categorized included 51 *Bsal* repeat families, 16 *Bd* families, 6 *Sp* families and 11 *Hp* families.

**Gene prediction, improvement and annotation for chytrids.** Gene prediction and annotation for *B. salamandrivorans* (*Bsal*), *B. dendrobatidis* (*Bd*), *S. punctatus* (*Sp*) and *H. polyrhiza* (*Hp*) was achieved using a *de novo* Eukaryotic Annotation Pipeline (Supplementary Table 2). The genome of *Hp* was downloaded from Genbank (accession AFSM00000000)[8] prior to annotation; as a gene set was not available in Genbank for this assembly, this assembly was annotated in parallel and that data is available on FigShare (https://dx.doi.org/10.6084/m9.figshare.4291310, https://dx.doi.org/10.6084/m9.figshare.4291313, https://dx.doi.org/10.6084/m9.figshare.4291283 and https://dx.doi.org/10.6084/m9.figshare.4291274). For *Sp*, the available assembly and annotation was used (Genbank BioProject PRJNA37881)[29]. RNA-Seq data was generated for *Bsal* and *Bd* and used to inform and update gene prediction, respectively. For *Bsal*, 78,103,411 paired-end reads (15.7 Terabases) of RNA was sequenced on three lanes of Illumina HiSeq from *in vitro* cultures (Supplementary Table 1). For *Bd*, initial gene calls were predicted using a combination of FGENESH[30], GENEID[31], EST-based genes and manual annotation; this initial set (previously made available on the Broad Institute website at https://www.broadinstitute.org/annotation/genome/batrachochytrium_dendrobatidis/MultiHome.html) was improved using 50,374,154 paired-end reads (10.1 Tb) from RNA sequenced on three lanes of Illumina HiSeq from *in vitro* cultures (Supplementary Table 1), as well as 1,390 transcripts (800 kb) from Bd infection (described in detail below). Updated Bd genes were given new locus ids starting with BDEG_20001. Trinity version r20140413p1 (ref. 32) was used to assemble the RNA-Seq with the genome-guided mode, minimum kmer = 2, and a max intron size of 10 kb. The programme to assemble spliced alignments[33] was then used to align these transcripts to the genome, which were subsequently filtered for repetitive sequence using Repbase[34], PFAM[35], TransposonPSI (TPSI)[36] and multiple similar mappings by BLAT[37].

All four genomes were annotated using TBLASTn, Genewise[38], tRNA, rfam, rnammer (rnaGenes)[39], GenemarkHmmEs, TPSI and Repbase[34]. We generated a training set using Genewise[38] and Genemark[38]. Next, GlimmerHmm[40], Snap[41] and Augustus[42] were used to generate *ab initio* gene models. The best gene model at a given locus was selected from these data sets using evidence modeller (EM)[43]; conserved genes missing in gene sets were identified with OrthoMCL[44] and combined with the EVM set. Genes were then filtered if >30% coding sequence overlapped TPSI[36] hits (*e*-value 1e − 10), overlapped repeat Pfam/TigrFam hits, or RepeatRunner[45] proteins.

For the updated *Bd* gene calls, the total number predicted remained similar to the previous non-RNA-Seq-guided set (150 dropped, no new genes added). However, the number of predicted transcripts was greatly increased (8,819→9,893), resulting from a large increase in those that are spliced (7,801→8,940), and only a modest decrease in those that are not spliced (1,018→953). Subsequently, there was a large increase in the total number of predicted exons (38,552→47,046) and introns (29,733→37,153), and >5% more of the genome was covered with genic and exonic regions. In addition, untranslated regions of transcripts that previously took up 1.16% of the *Bd* genome now covered 6.9% of the genome.

In total, 10,138 genes were identified for *Bsal* (5,291 on + strand, 4,847 on − strand), encoding 12,474 transcripts (9,875 spliced and 2,599 unspliced), encompassing 58,251 exons and 45,777 introns. *Bsal* has a lower proportion of spliced transcripts (~79%), compared with *Hp* (85%), *Sp* (88%) and *Bd* (90%). For *Hp* we identified a total of 6,254 genes encoding 6,254 transcripts. For *Sp* we identified 8,952 genes encoding 9,424 transcripts. *Sp* had an average of 6 exons per gene (56,727/8,952), which is greater than the average of 5 exons per gene in *Hp*, and 4.7 exons per gene in *Bd* and *Bsal*. All gene sets had high coverage of a set of 248 core eukaryotic genes, indicating that our annotation is highly complete; the updated RNA-Seq gene set was substantially improved for *Bd* (Supplementary Fig. 1a–c).

Genes were functionally annotated by assigning PFAM domains[35], GO terms, KEGG assignment and ortholog mapping to genes of known function. HMMER3 (ref. 46) was used to identify PFAM (release 27) and TIGRFam (release 12) domains, and BLASTx used against the KEGG v65 database (*e*-value < 1 × 10⁻¹⁰). GO terms were assigned using Blast2GO version2.3.5 (ref. 47), with a minimum *e*-value of 1 × 10⁻¹⁰. SignalP 4.0 (ref. 48) and TMHMM[49] were used to identify secreted proteins and trans-membrane proteins respectively (Supplementary Fig. 1d).

The protease composition of each chytrid was determined using top high scoring pairs from BLASTp searches (*e*-value < 1 × 10⁻⁵) made to the file 'pepunit.lib', which is a non-redundant library of 447,156 protein sequences of all the peptidases and peptidase inhibitors that are included in the MEROPS database (downloaded 2nd September 2014 from http://merops.sanger.ac.uk/). *Bd* had 586 top hits, *Bsal* had 589 total, *Sp* had 538 total and *Hp* had 416 total. M36 metalloproteases were aligned using MUSCLE v3.8.31 (ref. 50) and trimmed of excess gaps using trimAl 1.2rev59 (ref. 51) gappyout. We constructed the gene trees with RAxML v7.7.8 (ref. 52) and 1,000 bootstrap replicates, using the best-

fitting amino acid transition model (WAG) according to Bayesian information criterion implemented by Prottest v3.4 (ref. 53).

To explore CRN-like genes among the four chytrids, we downloaded the protein sequences from the *P. infestans* T30-4 genome (PRJNA17665), extracted the 437 annotated as Crinklers and ran BLASTp searches (*e*-values < 1e − 5) using that as our database. This identified 162 candidate CRNs for *Bd*, 10 for *Bsal*, 11 for *Hp* and 6 for *Sp* (162 in total). Of these, only 2 had signal peptides (*Hp* gene Hp_03611 and *Sp* gene SPPG_05862), suggesting few are secreted. Notably, 25 had Protein Kinase domains. Each of these genes were trimmed to 50 aa, and aligned using MUSCLE v3.8.31 (ref. 50). To identify subfamilies, we also BLASTp searched (*e*-values < 1e − 5) all of the proteins against the Crinkler protein domains[13]. In total, 141 predicted chytrid CRNs were assigned a known subfamily (119 for *Bd*, 9 for *Bsal*, 7 for *Hp*, 6 for *Sp*). CRN subfamilies did not clearly overlap with N-terminal domains.

We tested a variety of methods to optimize clustering of CRNs and to identify domains, including cd-hit v4.6-2012-04-25 (ref. 54) under a number of sequence similarity identities, as well as trimming the more divergent C-terminal to 35, 40, 45 and 50 aa, followed by, or proceeded by, a MUSCLE v3.8.31 alignment[50] with or without removing excess gaps using trimAl 1.2rev59 (ref. 51) gappyout. Motif searching was performed using GLAM2 (-Q -O. -M -z 20 -a 3 -b 15 -w 5 -r 30 -n 10000 -D 0.1 -E 2.0 -I 0.02 -J 1.0)[55]. Searching all of the sequences together after trimming to 50 aa did not yield a convincing single domain (Supplementary Fig. 7). Instead, we found that manually separating genes with two over-represented sequences obtained the highest confidence alignments spanning the most number of CRNs. The first domain (MPKR[YF]LNVEY) had a bit score of 1,576 and was present in 31 genes spanning most of the DAB subfamily (5/8), and half of the DXX subfamily (14/28) as well as 8/25 DXX-DHA, 2/5 DXX-newDXV. The second domain (YI[QK]YL) had a bit score of 1,191 and was present in 27 genes spanning all of the DFA-DDC subfamily and most of the DFB subfamily (12/13), as well as 1/2 DBE, 5/17 DX8, 1/28 DXX and 3/5 DXX-newDXV. The remaining 96 included a number of whole subfamilies such DN17 and newD2, and importantly, all of the *Bsal*, *Sp* and *Hp* predicted CRNs, but could not be subdivided into any convincing motifs (Supplementary Fig. 7).

To check if any novel chytrid CRNs were present, we constructed a Hidden Markov Model (HMM) using the two N-terminus sequences with HMMER 3.1b1 (ref. 46), and searched the protein sequences for the four chytrids with a *e*-value < 1 × 10⁻⁵ cutoff. The HMM from the first domain identified one new gene that had not previously been identified from *Bd* (BDEG_28597) and five DXX-DHA genes that had not been included in either of the domains. By now including them, we were able to obtain a higher bit score (1,576→1,772) for the new domain starting with R[YF]LNVEY (Supplementary Fig. 7). An updated HMM did not yield any further gene matches. Conversely, the HMM from the second domain identified 34 new genes that had not previously been identified from *Bd* and only one that had been identified and not included in the two domains (also not identified as in a CRN subfamily). Including these genes provided a large increase to the bit score (1,191→2,671) for a new domain starting with ME?[TN].{0,1}[KR].{0,1}FYI. Similarly, an updated HMM did not yield any further gene matches. The remaining 90 CRN's in neither domain 1 or 2 had a slightly reduced bit score (699→584) and only a low confidence motif: M.{0,39}[LV][EQ][ST]. A tree of all identified CRN 50aa N-termini was made using RAxML v7.7.8 (ref. 52) with 1,000 bootstrap replicates with the amino acid transition model WAG (Fig. 4a). Finally, intergenic distances (5′ and 3′) for each CRN were plotted (Supplementary Fig. 6), and distances were compared to full gene sets, secreted genes, and proteases for each species using *t*-tests (Supplementary Table 7).

To ascertain if the four chytrids secreted proteins included any large families (in addition to metalloproteases for example), we clustered all predicted secreted genes using MCL (http://micans.org/mcl/man/clmprotocols.html) with recommended settings '-I 1.4'. We clustered both the entire length of the protein (initially) and after cleaving the secreted peptide (Supplementary Fig. 3). Associated PFAM domains were found in all or nearly all members of some tribes, including the 2nd largest, which contained protease M36 domains, or the 6th largest which contained the peptidase S41 domain. The largest tribe had 105 proteins, and belonged entirely to *Bsal*, as did the 4th largest tribe. Many of the members of these secreted tribes were significantly differentially expressed between *in vivo* and *in vitro* conditions, including Tribe 1 (48%). Furthermore, these tribes are located almost exclusively in non-syntenic, unique regions of the *Bsal* genome (Fig. 1).

**Carbohydrate-binding module 18-containing genes.** Carbohydrate-binding proteins were identified with BLASTp searches to the Carbohydrate-Active enZYmes Database (http://www.cazy.org/) using the stringent *e*-value cutoff of e − 50 to avoid spurious hits. Looking at only the top high scoring pair, we identified very similar numbers between *Hp*, *Bsal*, *Bd* (*n* = 75, 87 and 93 respectively), and a greater number for *Sp* (*n* = 129). To examine carbohydrate binding modules (CBM), including CBM18, which is expanded in *Bd* and has been previously implicated in host–pathogen interactions[56], we downloaded the CBM18 protein domain family (Chitin_bind_1) HMM representation from the Sanger Pfam24.0 database[35], and used the HMMER3 application hmmsearch to search the four chytrid genomes with the same *e*-value cutoff of 0.01. In addition to the 18 previously identified CBM18 harbouring genes[56], we found three additional

CBM18 genes for *Bd*. We also increased the number of CBM18 domains found in *Bd* from 67 to 90. For *Sp*, we found just 5 genes with 8 domains. For *Bsal*, we found 15 genes with 30 domains. For *Hp* we found only 1 gene with 6 domains. None of the chytrid CBM18 genes were identified as a 1:1 ortholog. Each gene was aligned using muscle, and a tree was constructed with RAxML v7.7.8 (ref. 52) with 1,000 bootstrap replicates with the amino acid transition model WAG (Fig. 3).

Genes containing carbohydrate esterase 4 (CE4) superfamily mainly includes chitin deacetylases (EC 3.5.1.41) clustered together, and had been previously described as deacetylase-like (DE)[56]. *Bd* had 10 (previously 9 were reported[56]) although one lacked a DE domain. *Bsal* had 6 DE CBM18s, 1 of which lacked the DE domain. Sp had 3 DE CBM18s, and Hp had none. Another group of CBM18s contained a common central domain of tyrosinase, and previously described as tyrosinase-like (TL). *Bd* had 5 (previously 3 were reported[56]), *Bsal* had 3, *Hp* had 1 and *Sp* had none. The third group consisted of genes with no secondary domains and was previously described as lectin-like (LL). We recovered the same 6 genes in *Bd* as previously reported, and similarly found 6 in *Bsal*, 1 in *Sp* and none in *Hp*. However, the 6 *Bd* LL CBM18 had 48 CBM18 modules (averaging 8 per gene), while the 6 *Bsal* LL CBM18s had only 16 CBM18 modules (averaging 2.6 per gene). *Bsal* LL CBM18s are also considerably truncated compared with those of *Bd*, (mean *Bsal* protein length = 606, mean *Bd* protein length = 206). Finally, one of the CBM18 genes in *Sp* was drastically divergent from each of the others, and in addition to the CBM18 domain, also contained a glycoside hydrolase family 64 (GH64), a glycosyl hydrolase, family 18 (GH18), a Hormone receptor domain (HRM) and a Carbohydrate-binding (CB) domain. This gene is unlikely to be related in function to the other three groups, but could be used to root the tree.

Most *Bd* CBM18s are upregulated *in vivo*, 17/21 (81%), although largely nonsignificantly (Fig. 3). The four *Bd* CBM18s that are downregulated *in vivo* include three DE proteins (BDEG_23687, BDEG_23733 and BDEG_28751) and one TL protein (BDEG_26083). BDEG_23733 and BDEG_26083 are significantly downregulated *in vivo*. Three genes were significantly upregulated *in vivo*, including two DE proteins (BDEG_25495 and BDEG_25497) and one TL protein (BDEG_26084). Therefore, only the *Bd* LL are all upregulated *in vivo*, albeit nonsignificantly. Conversely, the *Bsal* truncated LL include three upregulated, one significantly (CBM18 07447) and three downregulated. Therefore, CBM18 genes differ among the chytrids both structurally and in expression. Only 7/15 (47%) *Bsal* CBM18s are upregulated *in vivo*. However, 5 of these are significantly upregulated including 2 TL, 2 DE and 1 LL (mentioned above). Interestingly, there is a phylogenetic signal to the regulation, whereby 1 of the 2 DE clusters is *Batrachochytrium* specific, and all genes contained are downregulated: 3 significantly (2/2 sig. downregulated *Bsal* genes + 1 *Bd* gene) and 2 nonsignificantly (termed downregulated DE; DRDE). The other DE cluster had *Sp* genes and *Batrachochytrium* genes: 2 of which were nonsignificantly downregulated, 3 were nonsignificantly upregulated and 4 were significantly upregulated.

**Ortholog-based phylogenetic and functional analysis of chytrids.** To reconstruct the evolutionary relationships between the four chytrid species, we identified 1:1 orthologs with OrthoMCL[44]. We aligned orthologs with MUSCLE v3.8.31 (ref. 50), extracted the CDS sequences in a codon context, and trimmed to the smallest contiguous sequence, and then concatenated alignments. In total, we aligned 4.1 Mb of transcripts for each chytrid. From greatest to smallest in nucleotide similarity of core orthologs, *Bsal* and *Bd* were 66% identical, *Bsal* and *Bd* were each 59% identical to *Hp* and *Sp* was 53% identical with *Bd*, *Bsal* and *Hp*. Prottest v3.4 (ref. 53) was used to determine the best-fitting amino acid transition model (DCMut) according to Bayesian information criterion. The final tree was produced using RAxML v7.7.8 (ref. 52) with 1,000 bootstrap replicates. DAGchainer[57] was used to identify maximally scoring syntenic blocks of two or more ordered gene pairs, which covered most of *Bd*, *Bsal* and *Hp* (72%, 50% and 87%, respectively), but only 14% of *Sp*, owing to its greater divergence.

We identified 3,378 genes (33% *Bsal* genes, 38% *Sp* genes, 39% *Bd* genes and 54% *Hp* genes) that were in 1:1 orthology between the four species (Supplementary Fig. 1c, Supplementary Table 3). Each chytrid had a large proportion of unique genes (*Hp* = 1,028 genes; 16%, *Bd* = 1,989 genes; 23%, *Bsal* = 3,300 genes; 33%, *Sp* = 3,343 genes; 37%). Unique *Hp* genes (1,028) belonged to 983 individual clusters (1.05:1), indicating a wide range of functions. Similarly, unique *Sp* genes (3,343) belonged to 2,884 clusters (1.16:1). Unique *Bd* and *Bsal* genes (1,989 and 3,300) belonged to far fewer clusters (935; 2.13:1 and 1,866; 1.77:1) indicating unique genes among the *Batrachochytrium* species are from larger expanded gene families. We found 3.58× the number of *Batrachochytrium*-specific genes (in *Bd* and *Bsal*) compared with those not found in *Bd* and *Bsal*.

Each chytrid was also uniquely missing a smaller proportion of gene orthogroups (*Hp* = 730; 12%, *Bd* = 160; 2%, *Bsal* = 95; 1%, *Sp* = 549; 6%). *Hp* had 1.3× more uniquely absent genes compared with unique genes. *Bd* and *Sp* both had >5× more unique genes than uniquely absent genes. Most strikingly, *Bsal* has 20× more unique genes compared with uniquely absent genes. This indicates that gene families among these four species are expanding overall, and the biggest expansions have occurred in the *Bsal* lineage.

The immense variation in genome size and gene content reflects both the selection and evolutionary differences between the species, but also the paucity of available chytrid genomes for comparison and the evolutionary history masked by that absence. Despite this sample-size limitation, the conservation of large

gene sets between the amphibian-infecting chytrids may harbour some of the more conserved pathogenicity-enabling genes. To explore this further, we performed two-tailed Fisher's exact test with *q*-value FDR for PFAM and GO terms in each of the orthogroups (Supplementary Table 3, Supplementary Fig. 1c) against the remaining orthogroups. Out of the 22,837 PFAM domains, we found 2,007 significantly enriched. Out of the 5,541 genes with GO terms, we found 959 were significantly enriched by comparing each orthogroup subset against the remaining gene sets. For *Bsal* specific genes, we found 14 enriched PFAM terms, the most significant of which are Protein kinase domain (5.87e − 44), Fungalysin metallopeptidase (3.71e − 42) and Protein tyrosine kinase (3.27e − 20). We found nine *Bd* specific enriched PFAM domains, the most significant of which are Eukaryotic aspartyl protease (4.73e − 86), Xylanase inhibitor C-terminal (2.5e − 66), Peptidase family S41 (3.55e − 12) and Fungalysin metallopeptidase (M36) (3.01e − 05). *Batrachochytrium*-specific genes (*Bd* + *Bsal*) included Polysaccharide deacetylase (3.12e − 15) and Chitin-recognition protein (7.22e − 07) enriched terms. Polysaccharide deacetylases such as chitin deacetylase act as virulence factors in other species such as *Cryptococcus neoformans*[58], as do chitin-recognition proteins[59].

Similar numbers of proteases were identified between *Sp*, *Bd* and *Bsal* (*n* = 538–589), and slightly fewer were found in *Hp* (*n* = 416). The protease profiles were, however, markedly different (Supplementary Fig. 5), with substantial increases of Aspartic peptidases A01 and A11 in *Bd*, and a large expansion (>3×) of M36 metalloproteases in *Bsal* (*n* = 110) compared with *Bd* (*n* = 35)—which is itself vastly expanded compared with *Hp* or *Sp* (*n* = 2 and 3, respectively) (Fig. 2). To test for non-classical secretion of proteins, including CRNs, we used SecretomeP[60] (NN-score ≥ 0.5) on each of the protein sequences of the four chytrids. However, most of the proteins were flagged as potentially being non-classically secreted (6,523 *Bsal* genes, 4,478 *Bd* genes, 4,581 *Sp* genes, 2,991 *Hp* genes), suggesting low-specificity. Cell wall genes were identified using OrthoMCL[44]-identified orthologs between *Bsal*, *Bd*, *Sp*, *Hp* and *Aspergillus nidulans* FGSC A4 (for which cell wall genes are well documented and more complete than those in *Saccharomyces cerevisiae*). For meiotic genes, we compared the chytrid orthologs to those of *S. cerevisiae* using OrthoMCL[44].

**Host–pathogen transcriptomes.** The animal experiment was performed with the approval of the ethical committee of the Faculty of Veterinary Medicine (Ghent University, EC2014/77) under strict BSL2 conditions.

First, the transcriptomal program of *Bd* and *Bsal* during *in vitro* cultures was determined. The fungi were cultured in PmTG broth and RNA from 5-day-old growing cultures was extracted with the RNeasy Plant kit (Qiagen), according to the manufacturer's instructions. Second, we sequenced the transcriptome of the salamander *T. wenxianensis* (*Tw*) during infection by *Bd* or *Bsal* or an uninfected control (Supplementary Table 1). Nine captive bred *Tw* were housed individually at 15 ± 1 °C on moist tissue, with access to a hiding place. All animals were clinically healthy and free of *Bd* and *Bsal* as assessed by sampling the skin using cotton-tipped swabs and subsequent performing qPCR. After 1 week of acclimatization, six animals were exposed to 1 ml of 10^4 *Bd* (three animals) or *Bsal* (ther animals) spores per ml water for 24 h[6]. Three additional animals were used as negative controls and were sham treated with 1 ml water for 24 h. Animals were fed twice weekly with crickets and followed up by clinical examination. At 10 days post infection (*Bd* infection load 1,100 GE per PCR reaction; *Bsal* infection load 1,900 GE per PCR reaction) the animals were killed and the skin was removed immediately. A part of the skin (10 mg) was stored in RNA later for 24 h and RNA then extracted with TRI Reagent combined with the RNeasy Plant kit (Qiagen). Skin samples for histopathology were stored in formalin. Histopathology confirmed the *Bd* or *Bsal* infection in all inoculated salamanders. Strand-specific libraries were constructed for each of the three replicates for each condition, using TruSeq RNA sample preparation with poly-A selection, and paired-end reads were generated on an Illumina HiSeq2000.

As no salamander gene set was available for comparison, we generated a new reference transcriptome assembly for analysis (Supplementary Table 8). We assembled *Tw* reads both from uninfected and infected samples with Trinity version r20140413p1 (ref. 32) using the strand-specific option (--SS_lib_type RF) and a minimum kmer coverage of 2 (--min_kmer_cov). We then aligned the resulting transcripts to the *Bsal* and *Bd* gene sets to remove fungal transcripts. Transdecoder was used to identify coding regions in these transcripts. After a final removal of *Batrachochytrium* transcripts, this Transcriptome Shotgun Assembly project was deposited at DDBJ/EMBL/GenBank under the accession GESS00000000. The version described in this paper is the first version, GESS01000000.

For each sample, we aligned reads to the transcripts of host or pathogen using Bowtie[61]. For *Bd* and *Bsal*, we aligned the reads to the RNA-Seq updated gene sets, and for *Tw* we aligned to the transdecoder coding regions. The RNA-Seq counts for biological replicate samples showed a high degree of correlation (Supplementary Fig. 8). We identified differential expression with our replicates using EdgeR[62], with significance set at FDR *P* value < 0.001 and > fourfold change of TMM normalized FPKM for the following five comparisons: (1) *Bd in vitro → Bd in vivo*, (2) *Bsal in vitro → Bsal in vivo*, (3) *Tw → Tw* with *Bd*, (4) *Tw → Tw* with *Bsal* and (5) *Tw* with *Bd → Tw* with *Bsal*. Two-tailed Fisher's exact test with *q*-value FDR for PFAM and GO terms in each of the differentially expressed

subsets were compared against the remaining set. For the enrichment tests, and identification of differentially expressed immune genes, we joined those genes that were significantly up or downregulated in *Bd* relative to *Bsal* with those up or downregulated in *Bd* relative to control, and did the same for *Bsal* (that is, merged up or down in *Bsal* relative to *Bd* with up or down in *Bsal* relative to control).

Overall, *Bd* genes appear downregulated *in vivo* (5,532/8,249; 67%; Supplementary Fig. 8d, Supplementary Data 1); however, this is probably affected by the lower sensitivity of the *in vivo* samples compared to *in vitro*. The majority of *Bd* G1M36s are upregulated *in vivo* (16/25; 64%) (Fig. 2b, five of which (20%) were significantly upregulated—while none were significantly downregulated *in vivo*. Surprisingly, 92% of the *Bd* CRNs were downregulated *in vivo*, with only 12 upregulated (2 DXX, 2 DX8, 1 DN17, 1 DX8, 1 newD2 and 5 belonging to no subfamily). Ten of these belonged to *Bd* CRN1, one of them to *Bd* CRN2 and the final one was in the remaining set.

*Bsal* genes were also mostly downregulated *in vivo*, but to a lesser extent than *Bd* (5,468/9,372; 56.5%; Supplementary Fig. 8b, Supplementary Data 1). Of these, 613 genes were significantly downregulated, while 447 were significantly upregulated *in vivo*. Much like *Bd*, 8/10 *Bsal* CRNs were downregulated, 3 of which were significantly downregulated. Similarly, the majority of *Bsal* G1M36's are upregulated *in vivo* (22/41; 54%). However, unlike *Bd*, eight (19%) were significantly downregulated *in vivo*, while 11 (27%) were significantly upregulated. This comparison suggests a more complex regulation of this subclass of protease in *Bsal*. *Bsal* G2M36s are mostly upregulated *in vivo* (26/36; 72%), 4 of which are significant—while none are significantly downregulated *in vivo*. A small number of M36s did not cluster with G1M36s or G2M36s, and included those also found in *Hp* and *Sp*. For *Bsal*, 3 of the 4 were downregulated *in vivo*, while 4 of the 5 were downregulated *in vivo* for *Bd*. Both of those upregulated *in vivo* were significant, and belonged to unique orthogroups.

We found substantial differences in host transcriptional response to *Bd* and *Bsal* (Supplementary Data 2). The largest category of genes differentially expressed in *Tw* were found from *Bd* infections. Most of these genes were upregulated ($n = 384$) compared with 96 for *Bsal*. *Tw* also had a large number of downregulated genes during *Bd* infection ($n = 106$) compared with only 12 for *Bsal*. Only two GO terms were enriched, including Immune system processes and extracellular region (Fisher exact test with *q*-value multiple correction), both from *Tw* genes upregulated during *Bd* infection ($q = 3.89e - 07$ and $5.15e - 06$, respectively) (Supplementary Data 3). We found that a large number of immune-related genes, including Antimicrobial peptides, B-cell related protein, interferons, macrophage proteins, MHCs, NF-KB and Toll-like receptors were differentially upregulated in *Bd*-infected salamanders (Fig. 5c, Supplementary Data 4), but with no differential expression found in *Bsal* infected animals. We also found a number of immune genes that were overwhelmingly differentially expressed by *Bd* infected animals, but also present in low numbers during *Bsal* infections (Fig. 5). These included cytokines, immunoglobulins, interleukins and Tumour Necrosis Factors (Supplementary Data 4). In addition to inflammatory factors, we saw a number of metalloproteinase inhibitors. Finally, a large number of transporters were differentially expressed during *Bd* infection, including nine downregulated and five upregulated aquaporins, which have been described as 'the plumbing system for cells'[63]. By comparison, only one (sodium/glucose cotransporter) was upregulated by animals infected by *Bsal*. Keratin was also differentially expressed, including three upregulated during *Bd* infection and 3 downregulated during *Bd* infection.

**Detection of protease activity and mRNA expression.** Per condition, ~150 mg of a mature *Bd* and *Bsal* culture containing all life stages was collected, centrifuged for 3 min at 3,000 r.p.m., and resuspended in unsupplemented 200 µl distilled water or 200 µl distilled water supplemented with antipain dihydrochloride (50 µg ml$^{-1}$), bestatin (40 µg ml$^{-1}$), chymostatin (60 µg ml$^{-1}$), E-64 (10 µg ml$^{-1}$), leupeptin (0.5 µg ml$^{-1}$), pepstatin (0.7 µg ml$^{-1}$), phosphoramidon (330 µg ml$^{-1}$), pefabloc (1 mg ml$^{-1}$), EDTA (0.5 µg ml$^{-1}$) or aprotinin (2 µg ml$^{-1}$). The cultures were lysed via sonication and the protease activity of the lysate was analysed using the Pierce Fluorescent Protease Assay Kit (Thermo Fisher Scientific), according to the manufacturer's instructions. Based on the results obtained, protease activity of *Bd* and *Bsal* spores ($10^7$ spores per condition) was examined in control spores and spores supplemented with antipain dihydrochloride (50 µg ml$^{-1}$), chymostatin (60 µg ml$^{-1}$), E-64 (10 µg ml$^{-1}$), pefabloc (1 mg ml$^{-1}$), EDTA (0.5 mg ml$^{-1}$) or a general protease inhibitor cocktail (Sigma-Aldrich), as described above. *Bd* and *Bsal* spores were collected by putting sterile distilled water on a culture flask containing mature sporangia. Once the zoospores were released, the water containing the zoospores was collected. To reduce the percentage of mature cells, the water containing the zoospores was passed over a sterile mesh filter with pore size 10 µm (Pluristrainer, PluriSelect). The flow through was used as the zoospore fraction (>90% purity). The results were analysed in SPPS using a non-parametric Mann–Whitney *U*-analysis.

Using TRI Reagent combined with the RNeasy plant kit (Qiagen), total RNA was isolated from freshly collected spores ($10^7$ spores per condition), spores ($10^7$ spores per condition) that were incubated with chytrid negative skin tissue of *T. wenxianensis* for 2 h in distilled water, 3 day old sporangia ($5 \times 10^6$ sporangia per condition) treated with unsupplemented TGhL broth or TGhL broth supplemented with chitinase (Sigma-Aldrich; 0.200 units) for 2 h and skin tissue from the chytrid-infected *T. wenxianensis* animals (10 mg per animal). Pure *Bd* and

*Bsal* sporangia conditions (>90% purity) were obtained by seeding the zoospore fraction in a six well at a concentration of $5 \times 10^6$ zoospores per well in TGhL broth. The spores were incubated during 3 days at 20 °C or 15 °C, respectively, until they reached the sporangia life stage. The RNA concentration was measured by absorbance at 260 nm using a nanodrop spectrophotometer and the quality of the RNA samples was checked using an Experion RNA StdSens Analysis kit (Bio-Rad). Total RNA (1 µg) was reverse transcribed to cDNA with the iScript cDNA synthesis kit (Bio-Rad). The housekeeping genes α-centractin, R6046, TEF1a and GAPDH were included as reference genes. The list of genes and sequences of the primers used for quantitative PCR analysis can be found in Supplementary Table 10. Real-time quantitative PCR reactions were run in duplicate and the reactions were performed in 10 µl volumes using the iQ SYBR Green Supermix (Bio-Rad). The experimental protocol for PCR (40 cycles) was performed on a CFX384 RT-PCR cycler (Bio-Rad). The results are shown as fold changes of mRNA expression relative to the mRNA expression levels in fresh spores. Fold changes were calculated using the cycle threshold ($\Delta\Delta C_T$) method (Livak and Schmittgen, 2001), and they were analysed in SPPS using a non-parametric Kruskal–Wallis analysis, followed by a Dunn–Bonferroni *post hoc* test (Supplementary Table 6).

**Data availability.** The data reported in this paper are outlined in Supplementary Information files and the raw sequences are deposited at GenBank under Bioproject PRJNA326253 for *B. dendrobatidis* and Bioprojects PRJNA326249 and PRJNA311566 for *B. salamandrivorans*. The genome assemblies and annotations are deposited at GenBank under Bioprojects PRJNA13653 for *B. dendrobatidis* and PRJNA311566 for *B. salamandrivorans*. The transcriptome assembly of *T. wenxianensis* is deposited at GenBank under accession GESS00000000, and RNA-Seq data from *T. wenxianensis* is deposited at GenBank under Bioproject PRJNA300849.

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

## Acknowledgements

We acknowledge the Broad Institute Sequencing Platform and Imperial College London for generating the DNA and RNA sequence described here. Financial support was provided by a UK Natural Environmental Research Council (NERC NE/K012509/1) grant to MCF, a Wellcome Trust Fellowship to RF, a Morris Animal Foundation grant to FP, and by the National Human Genome Research Institute grant number U54HG003067 to the Broad Institute. E.V. is supported by the Research Foundation Flanders (FWO grant 12E6616N).

## Author contributions

Designed the study: A.M., J.E.L., T.Y.J., F.P., M.C.F. and C.A.C. Analysed data: R.A.F., A.M., E.V., A.A., F.P. and C.A.C. Performed and interpreted experiments: A.M., E.V., R.D. and F.P. Wrote the paper: R.A.F., A.M, F.P., M.C.F. and C.A.C.

## Additional information

**Competing interests:** The authors declare no competing financial interests.

