## [Peer Review File · Nature Communications]

Reviewers' comments:

Reviewer #1 (Remarks to the Author):

This article reports on the newly sequenced genome of *Batrachochytrium salamandrivorans* (Bsal) in comparison with the previously sequenced genome of *Batrachochytrium dendrobatidis* (Bd). The genomes of both pathogens were compared with those of two nonpathogenic chytrid species, *Homolaphlyctis polyrhiza* (Hp) and *Spizellomyces punctatus* (Sp). It adds additional RNA-seq data for expression patterns of both pathogens growing in liquid culture and the skin response of a susceptible salamander species (*Tylototriton wenxianensis*) following infection by both pathogens. The central findings are that Bd and Bsal share a number of gene families distinct from those shared with the two nonpathogenic chytrids that have a saprophytic lifestyle rather than infecting the skin. The unique gene families that are enriched in Bd and Bsal include a diverse array of M36 metalloprotease family members and Carbohydrate Binding Module 18 domain proteins. However, Bsal contains a larger genome than the other chytrids with expansion of several novel gene sets.

The findings are novel. Publication of this information will provide a highly valuable resource for amphibian disease scientists in their efforts to understand the virulence characteristics of these unique and lethal fungal pathogens.

While much of this manuscript is very-well written, it needs and deserves further editing to correct some minor problems and explain the possible biological significance of some of the information reported in the main manuscript and abundant supplemental files. My specific concerns and suggestions are listed below.

Specific Criticism or Suggestions:

1. Lines 1-2. Title. Titles are important, and this title seems a bit too broad. A more accurate title would be "Evolutionary innovations underpin possible niche-specific infection strategies in the pathogenic chytrid fungi infecting amphibians".
2. Lines 25-38. Abstract. I have some level of discomfort with the term "virulence effectors" on line 32. Although it is likely that some of the unique sets of genes described for Bd and Bsal will turn out to be virulence factors, it is not yet clear that they are necessarily linked to host invasion or pathogenicity in a vertebrate host. For example, the metalloproteases are one of the examples of genes expanded in the pathogenic chytrids. Although neither Hp nor Sp appear to have as many metalloproteases (only 3 or 2, respectively in comparison with 110 in Bsal and 35 in Bd), it has been reported that another nonpathogenic chytrid, *Allomyces macrogynus*, has 31 metalloprotease gene family members (Joneson S., et al. 2011). It would be more accurate to state that "divergent infection strategies appear to be linked to radiations of lineage-specific factors". That is, delete the term "virulence effectors" here. On lines 36-38, it would be more correct to state, "The breadth and evolutionary novelty of these "candidate" or "possible" virulence effectors underscores the urgent need to halt the advance of pathogenic chytrids and prevent incipient loss of biodiversity".
3. Lines 41-43 of the introduction. The statement that "Colonization of the vertebrate host niche is a relatively recent event in microbial history..." seems overstated. It seems more

likely that microbes and vertebrate hosts have evolved together. I recommend deleting the first sentence of the introduction entirely.

4. Line 50. The term "host spectra" is used multiple times. "Spectra" tend to be more often used to describe light properties or physical properties rather than groups of biological species. Here it would be best to revise to "host species range". In other places (lines 100 and 114) "host group" or "host species group" would be a better choice of words.

5. Lines 62 and 63. It is stated that you sequenced the genomes of Bd and Bsal. Although I think I have read all of the methods thoroughly, I may have misunderstood. Did you sequence the genome of Bd again or simply use the sequence information previously done by the Broad Institute of the JEL 423 isolate. Please be clear here and on line 77 and in other places in the text. This is not a criticism of the significant amount of information that was newly developed for this manuscript, but it is a suggestion to avoid taking credit for re-sequencing if it was not done.

6. Line 94. Please replace "most" with "about half". That is, 55% and 47% are about half.

7. Lines 124-128. Referring to Figures 2c-d, you state that "Regulation and activity of the expanded proteases is complex and life-stage specific". Given that the zoospore transition to more mature stages can occur very rapidly, it is critical that the zoospores providing the material were highly purified to filter out more mature stages. I could find no information about the careful preparation of the zoospores. If they were not highly pure zoospores, it would be important to qualify your results to indicate the possible proportion of more mature cells in the preparations. If they were simply washed from agar plates without further purification, they could contain a fairly high percentage of maturing thalli (Fites J.S., et al., 2013). Thus the interpretation would be misleading.

8. Figures 2d, 3c, S8, show significance differences using letter designations. This method is very confusing. It is often unclear which group is used for comparison and which differences are statistically or biologically significant. Please consider representing the statistical comparisons in a different manner. Alternatively, this data could be presented in tabular form and the specific comparisons described more carefully.

9. In Fig. 3c, the legend for the histogram with small dots is missing. Although the legend indicates that the readout for "spores" is always set at "1", the bars don't seem to consistently reflect that fact.

10. In Figure 2d and 3c, the letter designations of the Bd proteases in the text of the figure legends don't match the legends within the figures.

11. Please consult a statistician about the acceptable significance levels when doing multiple comparisons in Figures 2d, 3c, and S8. It is my understanding that if you use the Dunn-Bonferroni corrections to avoid type 1 errors, the p-value for significance would need to be $p < 0.0125$ for 4 different comparisons,

12. Lines 134-146. There is no mention of the data reported in Fig. 3b in the main text. It seems to show a complex and inconsistent pattern of expression of the CBM18 proteins in vitro and in vivo. Please explain what this figure panel is intended to show or omit it.

13. Figure S2c,-d. Some of the lettering in these figures is too small for this reviewer to read. I assume the supplemental information will be available as a pdf and even at 300X magnification, I was unable to decipher the letters. Please modify the figures or make them separate figures.

14. Figure S3. I'm not sure this figure is needed. Some of the information reported in this figure is redundant with Figure 4a.

15. Lines 83-94 of Supplemental text. There is a description of ploidy analysis, depth of coverage, and allele-balance. The last sentence refers to Figure S2B. However, I think it should be Figure S2A. Was this a typographical error? In this whole section, the references to figure panels seemed unclear. Although Fig S2D is shown, it is not mentioned in the supplemental text.

16. In general, I found the supplementary text and figures to be difficult to follow and fully understand. When examining the supplemental figures, it was not always clear to me what each figure was intended to show. Perhaps this could be addressed by adding a final sentence to each supplemental figure legend that summarizes the content. The numbering of the supplemental figures should follow the order in which they were introduced in the main text. For example, the supplemental figures mentioned in the main text jump from Figure S1 on line 85 to Figures S7 and S8 in the next paragraph.

17. There did not seem to be a cohesive organization to the supplemental information. I would suggest that it be organized as follows:

a. Supplemental methods that follow the order of the data as it is presented in the main text. The methods seemed to be scattered about in the supplemental information.

b. Supplemental presentation of data or further discussion with additional subheadings. For example, I would have preferred a subsection labeled "Ploidy of Bsal in relation to other chytrids". Another subsection could have been labeled "Repetitive content of Bsal in relation to other chytrids".

c. In the subsection labeled "Gene prediction improvement and annotation for chytrids", it would make more sense to begin this section with the text from lines 189-215.

Reviewer #2 (Remarks to the Author):

Manuscript 102342_0 "Evolutionary innovations underpin niche-specific infection strategies across emerging pathogenic fungi" by Farrer et al.

A./B. Summary of key results, originality and interest

The manuscript by Farrer et al. describes the sequencing and analysis of the genomes of two *Batrachochytrium* species with different host specificity, and their comparative analysis with the genomes of two related non-pathogenic chytrid fungi. Furthermore, transcriptomes were generated from different stages of the two *Batrachochytrium* species either in vitro or during infection of an amphibian host, as well as transcriptomes from the infected and non-infected host. The transcriptomes were used to identify genes and gene families that might be associated with host infection.

The study is extremely interesting, because the findings indicate that despite their evolutionary relationship, the two *Batrachochytrium* species seem to have evolved independent means for host infection, which might also explain the different range of hosts that they can infect. The differences among the two pathogenic chytrids as well as between these and the non-pathogenic chytrids include expansion of several gene families in the *Batrachochytrium* species and differences at the level of expression. The accompanying analysis of the transcriptome of the infected vs. non-infected host show different reactions of the host immune system to infection with the two chytrids that might explain the

different morphological symptoms of the caused diseases.

Overall, the study is highly interesting to researchers working on emerging pathogens or fungal evolution, and provides an important source of target genes and processes to further analyze in these fungal species that are a major threat to global amphibian biodiversity.

C.-E., G.-H.: The methodology is mostly appropriate, appropriate references are cited in most cases, and the presentation is overall clear. Some points that could be improved are given below (point F).

F. Suggested improvements:

1. line 93: Table S8 is a Table with primers, not information on differently expressed genes.
2. lines 157-161: It might be possible to check if the repeat expansions of some of the families were recent by analyzing the similarity within each repeat group. Repeat groups that have recently expanded should have higher average similarities than groups that underwent a bout of expansion long ago and have diverged since then.
3. Figure S6 is difficult to understand, because it is not indicated which experiments the individual numbers refer to. It would be better if additional information about the type of experiment were given (e.g. "Bd in vitro" etc., similar what is given in Table S1 for some of the experiments).
4. Lines 212-219: It might be somewhat premature to label the expanded/expressed genes as "arsenal of virulence factors" in the discussion, because even though they are associated with virulence and are certainly good candidates for future analysis, the genes were not yet shown to be causative to virulence, therefore it might be better to call them "putative virulence factors" or something alike.
5. There are prior publications of *Batrachochytrium* genomes and transcriptomes of different strains/conditions, even though the transcriptome analyses were not performed in vivo on a host (Rosenblum et al. PNAS 2008, PNAS 2013). Nevertheless, it would be important to know how the analyzed strains/conditions differ or are extended by the present study, and if the genomes of the analyzed strains differ from the ones analyzed here, especially as there seems to occur rapid genome evolution in this pathogenic genus. This could be included in the results or discussion.
6. There is only one accession number given for the transcriptome of the amphibian host, but no accession numbers for the genome reads, transcriptome reads, and genome assemblies/annotation of the analyzed fungi. The corresponding accession numbers should be given (e.g. for databases like SRA, GEO, GenBank) to make these valuable resources accessible to the scientific community. A good place to include them might be Table S1.
7. Figures with phylogenetic trees: The unit of the scale bar is not mentioned in the Figure legends.

8. Figure 2A: It would be nice to have a Table with the M36 proteins that were used for the analysis indicating which protein falls into which branch of the tree somewhere in the Supplement.

9. Tylototriton is misspelled in lines 89 and 468.

Reviewer #3 (Remarks to the Author):

Farrer et al. present a novel contribution to the study of two globally important fungal amphibian pathogens *Batrachochytrium dendrobatidis* (Bd) and *Batrachochytrium salamandrivorans* (Bsal). They present full genome sequences for both pathogens, offering the first published full genome of Bsal. Accompanying these sequences is a detailed comparison of gene families between the two pathogens and two of their closest fungal relatives. The authors compared gene expression of both pathogens in vitro and in vivo – after inoculation of one susceptible salamander species *Tylototriton wenxianensis* (Tw). They also offer gene expression analysis for Tw in response to both pathogens, highlighting some intriguing differences in host response. While many of these findings are important contributions to the study of this disease system, there are places where data feel over interpreted and caveats are missing. Some general concerns are detailed below and specific comments on line numbers follow.

General comments:

The authors use language throughout that implies evolutionary causality. Only two pathogenic lineages are studied and they are each other's closest known relatives. Therefore, it is nearly impossible to make conclusions about the directionality of evolution or to convincingly assert cause and effect. Throughout the manuscript, it is important to be clear that certain genomic patterns are **associated** with difference between Bd and Bsal, but it is impossible to say that those genomic patterns **caused** observed differences between the species. Moreover, without additional species and explicit ancestral reconstruction inferences, the authors need to moderate statements that assume directionality in evolution.

The authors studied Bd, Bsal, and host gene expression in **one** amphibian host species. This provides an important in vivo comparison for the study, but sweeping generalizations about host responses to each pathogen cannot be made. Similarly the authors studied **one** Bd isolate, although there is well-documented diversity in Bd in both genotype and phenotype. The writing implies that the findings are generalizable across Bd, while this may not be the case. The study design is appropriate for the comparative framework (i.e., it would be beyond the scope of a single paper to do this study in a large number of isolates and host species). However, it is important to be clear about the need for further work to establish increased generality. Again, toning down language and adding explicit caveats about the limitations of analyzing one representative isolate and one representative host species is important.

The figures are over-complicated and difficult to interpret. The visual communication of information needs to be revisited and figures should be more parallel in structure and simplified.

Line Comments:

Line 1: The title "Evolutionary innovations underpin niche-specific infection strategies across emerging pathogenic fungi" seems inappropriate at several levels. There are important and exciting genomic patterns associated with different infection strategies in two pathogenic species, but the title suggests more causality, more directionality of evolution and a more comprehensive sampling across emerging pathogenic fungi that is not well supported by the data presented.

Line 33-35: "evolutionary traits that underpin a strikingly different immune response within a shared host species..." The data show that Bd and Bsal exhibit different expression patterns in one common host species, so "evolutionary traits" and "underpin" seem to oversell the data.

Line 54: "acquisition of fungal pathogenicity that allowed infection of vertebrates followed by evolution of host specificity". I am not convinced that we know the direction of evolution - and this has important implications here and other places in the manuscript. Since their divergence, Bsal could have evolved increased host specificity, Bd could have evolved increased generality, or both.

Line 72: There is no mention of sequencing the Tw transcriptome in the Introduction, which seems odd because much of the discussion relies on the findings from that experiment.

Line 75: I am not convinced that the focus on "evolutionary adaptation" here and throughout is necessary. The authors have demonstrated important genomic changes, but not necessary adaptation per se (we don't know if these genomic patterns were driven by natural selection).

Line 81: Is it true that not only is the Bsal genome bigger than the Bd genome, but the Bd genome is also bigger than the other chytrids studied? It would be good to report in the main text the Bd genome size (is it 8,952?).

Line 116: The terms "Amphibian/Caudate Destroying M36s" are unnecessarily overdramatic. The authors have not unambiguously determined the functional effect of these genes and gene families. "Associated" instead of "Destroying" would be much more appropriate.

Line 130: Example of place where the word "demonstrates" seems to overstep the data, "suggests" would be better.

Line 205-207: The findings are presented in a way that implies generality (e.g., that Bsal has immune dampening properties across caudates). The conclusions they can make from

one host species are limited and this hypothesis will have to be tested in a broader range of host species.

Line 208: There should either be a new paragraph added at the end of the Results or Discussion section that addresses necessary caveats, or caveats should be integrated throughout the existing Results section in appropriate places (e.g., the need for ongoing studies in other host species, with other Bd isolates, possible biases with genome sequencing methods, etc.)

Line 211: The claim that the expansion of virulence factors "has expanded the host range..." is assigning causality and directionality to evolution that cannot be supported by the data. Again, I would like to see a clear distinction between association and causality. For example, this could be modified to: "differential expansion of putative virulence factors by at least two chytrid fungi is associated with an expanded host range..."

Line 214-220: This paragraph uses unnecessary dramatic and somewhat redundant language that oversells the findings. Examples include "impressively" "arsenal" and "armamentarium".

Line 234: This is the first mention of which strain of Bd is used in this study. This should be presented earlier and the choice of strain/acknowledgement of the diversity of Bd should be mentioned.

Line 229-243: Bsal was sequenced with short reads and short inserts (<2.5kb) whereas Bd was sequenced with Sanger technology with 4kb, 10kb and 40 kb fosmids. I am concerned about whether there could be any artifacts that could arise from using two different methods for sequencing each genome. I would like to see a brief discussion about whether some genomic elements may be differentially represented in the assembly based on the sequencing approach.

Line 318, 321: "a transcriptome" would be more appropriate than "the transcriptome" given that this the sequencing is from a limited set of tissues under particular conditions.

Line 358-378: I was quite confused about how zoospores were separated from sporangia. Line 358 states that all life stages were collected, but then line 375 talks about separate samples of spores vs sporangia. Please clarify methodological details and specific how pure the samples of each life stage are expected to be.

Figure 1: Why is the figure arranged so that Sp is being compared to Bsal? It seems like it would be more phylogenetically informative and accurate to rotate the tree around that node so that Sp is compared to Hp.

Figure 2-4: Figure 2, 3 and 4 are extremely crowded, confusing, and difficult to interpret. The most essential information should be conveyed in very parallel structure across figures. Below are just some of the questions I had, but I think the structure and content of these figures should be rethought from the ground up.

Figure 2: What do the trees on top and to the side of the heat map represent? They should be either explained in the caption or removed. The labels to the right of the heat map are unreadable and could be simplified to just "Bd" or "Bsal." The figures in c and d should be simplified or removed.

Figure 3: The representation of the gene structure seems unnecessary and makes the figure confusing. In general, there should be more of a parallel structure between the gene family heatmap figures to make all of them more interpretable (i.e., trees vs networks). The figures in c are hard to interpret.

Figure 4: The Bd CRN type letter plots seem unnecessary. Same comment about the bar graphs in figure c – they should be made more interpretable or removed to simplify the display.

The supplemental materials are long and complicated.

Point by point response to reviewer's comments

Reviewer #1 (Remarks to the Author):

This article reports on the newly sequenced genome of *Batrachochytrium salamandrivorans* (Bsal) in comparison with the previously sequenced genome of *Batrachochytrium dendrobatidis* (Bd). The genomes of both pathogens were compared with those of two nonpathogenic chytrid species, *Homolaphlyctis polyrhiza* (Hp) and *Spizellomyces punctatus* (Sp). It adds additional RNA-seq data for expression patterns of both pathogens growing in liquid culture and the skin response of a susceptible salamander species (*Tylotriton wenxianensis*) following infection by both pathogens. The central findings are that Bd and Bsal share a number of gene families distinct from those shared with the two nonpathogenic chytrids that have a saprophytic lifestyle rather than infecting the skin. The unique gene families that are enriched in Bd and Bsal include a diverse array of M36 metalloprotease family members and Carbohydrate Binding Module 18 domain proteins. However, Bsal contains a larger genome than the other chytrids with expansion of several novel gene sets.

The findings are novel. Publication of this information will provide a highly valuable resource for amphibian disease scientists in their efforts to understand the virulence characteristics of these unique and lethal fungal pathogens.

While much of this manuscript is very-well written, it needs and deserves further editing to correct some minor problems and explain the possible biological significance of some of the information reported in the main manuscript and abundant supplemental files. My specific concerns and suggestions are listed below.

Specific Criticism or Suggestions:

1. Lines 1-2. Title. Titles are important, and this title seems a bit too broad. A more accurate title would be “Evolutionary innovations underpin possible niche-specific infection strategies in the pathogenic chytrid fungi infecting amphibians”.

A) We thank the reviewer for these comments. We report on a number of evolutionary innovations that are linked to differing phenotypes. The qualification of ‘possible’ does not seem necessary in the title, given that evolutionary innovation is overwhelmingly likely for differences in infection strategies, given the controlled infections reported in this paper. However, we have clarified that it is in ‘chytrid’ fungi as suggested.

2. Lines 25-38. Abstract. I have some level of discomfort with the term “virulence effectors” on line 32. Although it is likely that some of the unique sets of genes described for Bd and Bsal will turn out to be virulence factors, it is not yet clear that they are necessarily linked to host invasion or pathogenicity in a vertebrate host. For example, the metalloproteases are one of the examples of genes

expanded in the pathogenic chytrids. Although neither Hp nor Sp appear to have as many metalloproteases (only 3 or 2, respectively in comparison with 110 in Bsal and 35 in Bd), it has been reported that another nonpathogenic chytrid, *Allomyces macrogynus*, has 31 metalloprotease gene family members (Joneson S., et al. 2011). It would be more accurate to state that “divergent infection strategies appear to be linked to radiations of lineage-specific factors”. That is, delete the term “virulence effectors” here.

A) We agree that the term “virulence effector” is not directly supported, and have replaced with “factor” as suggested. It is indeed curious that a non-pathogenic chytrid should also harbor an independent expansion of M36 proteases. And we have included a comment to this effect in the discussion. However, replacing “linked” with “appear to be linked” is too equivocal, as it is inevitable that the breadth of changes that we have documented will cause divergent host-pathogen interactions.

On lines 36-38, it would be more correct to state, “The breadth and evolutionary novelty of these “candidate” or “possible” virulence effectors underscores the urgent need to halt the advance of pathogenic chytrids and prevent incipient loss of biodiversity”.

A) Agreed. We have updated the text accordingly.

3. Lines 41-43 of the introduction. The statement that “Colonization of the vertebrate host niche is a relatively recent event in microbial history...” seems overstated. It seems more likely that microbes and vertebrate hosts have evolved together. I recommend deleting the first sentence of the introduction entirely.

A) We agree that this sentence is a bit overboard – although obviously the origin of vertebrates is more recent than the origin of microbes. We have changed this to discuss the ongoing processes of host-range expansions.

4. Line 50. The term “host spectra” is used multiple times. “Spectra” tend to be more often used to describe light properties or physical properties rather than groups of biological species. Here it would be best to revise to “host species range”. In other places (lines 100 and 114) “host group” or “host species group” would be a better choice of words.

A) We have made the suggested changes.

5. Lines 62 and 63. It is stated that you sequenced the genomes of Bd and Bsal. Although I think I have read all of the methods thoroughly, I may have misunderstood. Did you sequence the genome of Bd again or simply use the sequence information previously done by the Broad Institute of the JEL 423 isolate. Please be clear here and on line 77 and in other places in the text. This is not a criticism of the significant amount of information that was newly developed

for this manuscript, but it is a suggestion to avoid taking credit for re-sequencing if it was not done.

- A) This is the first publication to report the genome (and transcripts) of the JEL423 isolate of *Bd*, sequenced as noted by the reviewer by the Broad Institute, and all the contributors to that genome project are authors here. There was an initial assembly and annotation that were publicly released. For this paper, we did not re-assemble the genome of JEL423; however to improve the gene set used here for analysis, we re-annotated the JEL423 genome using RNA-Seq. We also describe the assembly and annotation of the *Bsal* genome.

6. Line 94. Please replace “most” with “about half”. That is, 55% and 47% are about half.

- A) Changed as suggested.

7. Lines 124-128. Referring to Figures 2c-d, you state that “Regulation and activity of the expanded proteases is complex and life-stage specific”. Given that the zoospore transition to more mature stages can occur very rapidly, it is critical that the zoospores providing the material were highly purified to filter out more mature stages. I could find no information about the careful preparation of the zoospores. If they were not highly pure zoospores, it would be important to qualify your results to indicate the possible proportion of more mature cells in the preparations. If they were simply washed from agar plates without further purification, they could contain a fairly high percentage of maturing thalli (Fites J.S., et al., 2013). Thus the interpretation would be misleading.

- A) Spores for both *Bd* as *Bsal* were collected by putting fresh water on a culture flask containing mature sporangia. Once the zoospores were released, the water containing the zoospores was collected. In order to reduce the percentage of mature cells, the water containing zoospores was passed over a sterile mesh filter with pore size 10 µm (PluriStrainer, PluriSelect). This information has been added to the Supplemental info lines 512-516.

8. Figures 2d, 3c, S8, show significance differences using letter designations. This method is very confusing. It is often unclear which group is used for comparison and which differences are statistically or biologically significant. Please consider representing the statistical comparisons in a different manner. Alternatively, this data could be presented in tabular form and the specific comparisons described more carefully.

- A) We have opted to show this additional information in an additional supplemental table rather than showing it in the figures as suggested.

9. In Fig. 3c, the legend for the histogram with small dots is missing. Although the legend indicates that the readout for “spores” is always set at “1”, the bars don’t seem to consistently reflect that fact.

A) The figure legend states “which is considered 1”. Using the cycle threshold ($\Delta\Delta C_T$) method (Livak and Schmittgen, 2001), the mean value of the control group should be around 1. It is only exactly 1 if all the repeats for the spore fraction are exactly the same and give an exact same ct-value, which is of course never the case. Given a small variation between the repeats, the value will be close to 1 and will have a small standard deviation. The larger the variation between the repeats, the larger the value of the control group/standard deviation.

10. In Figure 2d and 3c, the letter designations of the Bd proteases in the text of the figure legends don’t match the legends within the figures.

A) This has been corrected.

11. Please consult a statistician about the acceptable significance levels when doing multiple comparisons in Figures 2d, 3c, and S8. It is my understanding that if you use the Dunn-Bonferroni corrections to avoid type 1 errors, the p-value for significance would need to be $p < 0.0125$ for 4 different comparisons,

A) Since our data are not normally distributed, we are restricted to using a non-parametric test, namely a Kruskal Wallis analysis. This test gives you an idea whether you have a significant difference between “one of the groups”. If you want to know between which groups there’s a significant difference, you can:

- 1) Perform a Kruskal Wallis analysis followed by pairwise Mann-Whitney U comparisons. Then you have to manually adapt your significance level to 0.0125 for 4 different comparisons (Bonferroni correction).
- 2) Perform a Kruskal Wallis followed by a post-hoc test of the Kruskal-Wallis, namely Dunn-Bonferroni post hoc test (Dunn 1964). The SPSS statistics software includes a Dunn-Bonferroni post hoc method following a significant Kruskal-Wallis test in the NPTEST procedure. This post-hoc test automatically analyzes the pairwise comparisons between the different groups, if the Kruskal-Wallis test was significant and gives you an adjusted p-value by multiplying the p value with the number of pairwise comparisons. This means that the software keeps the level of significance at 0.05, but corrects the p-values for the pairwise comparisons.

Example of output:

Each node shows the sample average rank of groep.

Sample1-Sample2	Test Statistic	Std. Error	Std. Test Statistic	Sig.	Adj.Sig.
3,00-2,00	6,167	4,082	1,511	,131	,785
3,00-1,00	14,167	4,082	3,470	,001	,003
3,00-4,00	-15,667	4,082	-3,838	,000	,001
2,00-1,00	8,000	4,082	1,960	,050	,300
2,00-4,00	-9,500	4,082	-2,327	,020	,120
1,00-4,00	-1,500	4,082	-,367	,713	1,000

Each row tests the null hypothesis that the Sample 1 and Sample 2 distributions are the same. Asymptotic significances (2-sided tests) are displayed. The significance level is ,05.

Both approaches can be used, but the first approach is a more conservative approach. Nevertheless, the results of both approaches are the same.

12. Lines 134-146. There is no mention of the data reported in Fig. 3b in the main text. It seems to show a complex and inconsistent pattern of expression of the CBM18 proteins in vitro and in vivo. Please explain what this figure panel is intended to show or omit it.

A) Fig. 3B indeed does show exactly that – a complex pattern of expression. Originally, this heatmap panel (like the others) was intended to show the expression patterns of key gene families that we had highlighted in the text, even in this case where there is no simple pattern. However, given this comment, we have omitted this panel and clarified in the text that no significant changes in expression between conditions were identified.

13. Figure S2c,-d. Some of the lettering in these figures is too small for this reviewer to read. I assume the supplemental information will be available as a pdf and even at 300X magnification, I was unable to decipher the letters. Please modify the figures or make them separate figures.

A) The font size was increased in parts C-D of figure S2 to improve readability as suggested.

14. Figure S3. I'm not sure this figure is needed. Some of the information reported in this figure is redundant with Figure 4a.

- A) Figure S3 details our efforts to characterize the *Batrachochytrium* CRN motif, both with initial multiple alignments, and following the use of a HMM to reveal additional putative CRNs. While 2/8 motifs shows (the final ones) do appear in a main figure, we feel this information is suitable in both places – and overall will be valuable to anyone interested in this class of genes, and the outcome of these analyses.

15. Lines 83-94 of Supplemental text. There is a description of ploidy analysis, depth of coverage, and allele-balance. The last sentence refers to Figure S2B. However, I think it should be Figure S2A. Was this a typographical error? In this whole section, the references to figure panels seemed unclear. Although Fig S2D is shown, it is not mentioned in the supplemental text.

- A) The description of chytrid heterozygosity and ploidy begins from line 67 in the supplementary text. This first paragraph deals exclusively with part S2A (which has been reordered according to separate comments, so is now S5a). The paragraph starting on line 83 described phasing analysis– which further indicated diploidy. Finally, we looked at how *BsaI* ploidy overlapped with syntenic regions (Figure S5B) – which included an initial removal of all contigs less than 5 kb. Therefore, we did indeed intend to refer to S5B. A reference to part S5D was added to the text.

16. In general, I found the supplementary text and figures to be difficult to follow and fully understand. When examining the supplemental figures, it was not always clear to me what each figure was intended to show. Perhaps this could be addressed by adding a final sentence to each supplemental figure legend that summarizes the content.

- A) The supplementary text and figures cover a substantial amount of different analyses, which we understand may need to be better anchored in the paper. As suggested, we have provided a brief statement of the meaning of each figure as the first sentence of each legend, highlighted in bold. We have also made some adjustments aimed to improve the clarity of each supplemental figure.

The numbering of the supplemental figures should follow the order in which they were introduced in the main text. For example, the supplemental figures mentioned in the main text jump from Figure S1 on line 85 to Figures S7 and S8 in the next paragraph.

A) Previously we used the order that the figures were covered in the supplemental text, rather than the main manuscript – so we have rearranged to match the main text as suggested.

17. There did not seem to be a cohesive organization to the supplemental information. I would suggest that it be organized as follows:

- a. Supplemental methods that follow the order of the data as it is presented in the main text. The methods seemed to be scattered about in the supplemental information.
- b. Supplemental presentation of data or further discussion with additional subheadings. For example, I would have preferred a subsection labeled “Ploidy of Bsal in relation to other chytrids”. Another subsection could have been labeled “Repetitive content of Bsal in relation to other chytrids”.
- c. In the subsection labeled “Gene prediction improvement and annotation for chytrids”, it would make more sense to begin this section with the text from lines 189-215.

A) These two subsections were part of a single “genome sequencing” section; these are now separated into the new subsections as suggested.

Reviewer #2 (Remarks to the Author):

Manuscript 102342_0 "Evolutionary innovations underpin niche-specific infection strategies across emerging pathogenic fungi" by Farrer et al.

A./B. Summary of key results, originality and interest

The manuscript by Farrer et al. describes the sequencing and analysis of the genomes of two *Batrachochytrium* species with different host specificity, and their comparative analysis with the genomes of two related non-pathogenic chytrid fungi. Furthermore, transcriptomes were generated from different stages of the two *Batrachochytrium* species either in vitro or during infection of an amphibian host, as well as transcriptomes from the infected and non-infected host. The transcriptomes were used to identify genes and gene families that might be associated with host infection.

The study is extremely interesting, because the findings indicate that despite their evolutionary relationship, the two *Batrachochytrium* species seem to have evolved independent means for host infection, which might also explain the different range of hosts that they can infect. The differences among the two pathogenic chytrids as well as between these and the non-pathogenic chytrids include expansion of several gene families in the *Batrachochytrium* species and differences at the level of expression. The accompanying analysis of the transcriptome of the infected vs. non-infected host show different reactions of the host immune system to infection with the two chytrids that might explain the different morphological symptoms of the caused diseases.

Overall, the study is highly interesting to researchers working on emerging

pathogens or fungal evolution, and provides an important source of target genes and processes to further analyze in these fungal species that are a major threat to global amphibian biodiversity.

C.-E., G.-H.: The methodology is mostly appropriate, appropriate references are cited in most cases, and the presentation is overall clear. Some points that could be improved are given below (point F).

F. Suggested improvements:

1. line 93: Table S8 is a Table with primers, not information on differently expressed genes.

A) Fixed – this should have been Table S7.

2. lines 157-161: It might be possible to check if the repeat expansions of some of the families were recent by analyzing the similarity within each repeat group. Repeat groups that have recently expanded should have higher average similarities than groups that underwent a bout of expansion long ago and have diverged since then.

A) It is true that it might be possible to look into sequence divergence between repeat families, and get a further insight into their evolution. While this is certainly an interesting avenue of research, we feel that this is beyond the scope of this paper, which as mentioned, already includes a vast supplementary section of supporting analyses. Further study of the repeat expansion including dating of different classes of elements would be better achieved in a dedicated follow-up paper.

3. Figure S6 is difficult to understand, because it is not indicated which experiments the individual numbers refer to. It would be better if additional information about the type of experiment were given (e.g. "Bd in vitro" etc., similar what is given in Table S1 for some of the experiments).

A) We have replaced the names with a more informative description of the conditions from which they were generated.

4. Lines 212-219: It might be somewhat premature to label the expanded/expressed genes as "arsenal of virulence factors" in the discussion, because even though they are associated with virulence and are certainly good candidates for future analysis, the genes were not yet shown to be causative to virulence, therefore it might be better to call them "putative virulence factors" or something alike.

A) We have qualified these as putative as suggested.

5. There are prior publications of *Batrachochytrium* genomes and transcriptomes of different strains/conditions, even though the transcriptome analyses were not performed in vivo on a host (Rosenblum et al. PNAS 2008, PNAS 2013). Nevertheless, it would be important to know how the analyzed strains/conditions differ or are extended by the present study, and if the genomes of the analyzed strains differ from the ones analyzed here, especially as there seems to occur rapid genome evolution in this pathogenic genus. This could be included in the results or discussion.

A) As noted by the reviewer, these previous studies fundamentally differ in only looking at gene expression in vitro. As our study also includes such samples, it is unclear what would be gained from comparing to additional in vitro samples that would introduce additional variable such as different strains. However, we agree that a comparison of gene expression between different strains could be an interesting future study, particularly where those strains show different phenotypes of some sort.

6. There is only one accession number given for the transcriptome of the amphibian host, but no accession numbers for the genome reads, transcriptome reads, and genome assemblies/annotation of the analyzed fungi. The corresponding accession numbers should be given (e.g. for databases like SRA, GEO, GenBank) to make these valuable resources accessible to the scientific community. A good place to include them might be Table S1.

A) We agree that many of these datasets will be of interest to the community. We have included NCBI bioproject identifiers for each of the datasets in the acknowledgments, and these contain links to all the data including the SRA. We have also included SRA links for all the data in Table S1 as suggested.

7. Figures with phylogenetic trees: The unit of the scale bar is not mentioned in the Figure legends.

A) The unit of scale is the mean number of nucleotide substitutions per site. We have added this information into the legends of the relevant figures.

8. Figure 2A: It would be nice to have a Table with the M36 proteins that were used for the analysis indicating which protein falls into which branch of the tree somewhere in the Supplement.

A) We have created a new supplemental table with this information.

9. Tylotriton is misspelled in lines 89 and 468.

A) Fixed.

Reviewer #3 (Remarks to the Author):

Farrer et al. present a novel contribution to the study of two globally important fungal amphibian pathogens *Batrachochytrium dendrobatidis* (Bd) and *Batrachochytrium salamandrivorans* (Bsal). They present full genome sequences for both pathogens, offering the first published full genome of Bsal. Accompanying these sequences is a detailed comparison of gene families between the two pathogens and two of their closest fungal relatives. The authors compared gene expression of both pathogens in vitro and in vivo – after inoculation of one susceptible salamander species *Tylotriton wenxianensis* (Tw). They also offer gene expression analysis for Tw in response to both pathogens, highlighting some intriguing differences in host response. While many of these findings are important contributions to the study of this disease system, there are places where data feel over interpreted and caveats are missing. Some general concerns are detailed below and specific comments on line numbers follow.

General comments:

The authors use language throughout that implies evolutionary causality. Only two pathogenic lineages are studied and they are each other's closest known relatives. Therefore, it is nearly impossible to make conclusions about the directionality of evolution or to convincingly assert cause and effect. Throughout the manuscript, it is important to be clear that certain genomic patterns are **associated** with difference between Bd and Bsal, but it is impossible to say that those genomic patterns **caused** observed differences between the species. Moreover, without additional species and explicit ancestral reconstruction inferences, the authors need to moderate statements that assume directionality in evolution.

A) We agree and now make it clear throughout that the differences that we observe are associated and that further research is needed to prove causality.

The authors studied Bd, Bsal, and host gene expression in **one** amphibian host species. This provides an important in vivo comparison for the study, but sweeping generalizations about host responses to each pathogen cannot be made. Similarly the authors studied **one** Bd isolate, although there is well-documented diversity in Bd in both genotype and phenotype. The writing implies that the findings are generalizable across Bd, while this may not be the case. The study design is appropriate for the comparative framework (i.e., it would be beyond the scope of a single paper to do this study in a large number of isolates

and host species). However, it is important to be clear about the need for further work to establish increased generality. Again, toning down language and adding explicit caveats about the limitations of analyzing one representative isolate and one representative host species is important.

A) We have now included an additional paragraph to the discussion on the caveats and potential future work from this study. We have toned down the language in numerous places to reflect more clearly the work that has been done.

The figures are over-complicated and difficult to interpret. The visual communication of information needs to be revisited and figures should be more parallel in structure and simplified.

A) We have trimmed down some of the figure panels as previous reviewers have requested, and also moved some of the statistical analysis from a number of the main figures to a separate supplemental table.

Line Comments:

Line 1: The title "Evolutionary innovations underpin niche-specific infection strategies across emerging pathogenic fungi" seems inappropriate at several levels. There are important and exciting genomic patterns associated with different infection strategies in two pathogenic species, but the title suggests more causality, more directionality of evolution and a more comprehensive sampling across emerging pathogenic fungi that is not well supported by the data presented.

A) We have not implicitly mentioned causality or directionality of evolution. Evolution innovates, and in this instance, resulted in two pathogens with strikingly different genotypic and phenotypic traits, with the former underpinning the latter. In terms of sampling, an $n=2$ still warrants the plural fungi. We do not think that clarifying "two pathogenic fungi" is necessary in the title, given that it is accurate as it is. We have however clarified that the paper is on chytrid fungi only.

Line 33-35: "evolutionary traits that underpin a strikingly different immune response within a shared host species..." The data show that *Bd* and *Bsal* exhibit different expression patterns in one common host species, so "evolutionary traits" and "underpin" seem to oversell the data.

A) *Bd* and *Bsal* are separated by millions of years of divergence, resulting in a huge difference in genome architecture and gene families, as well as "different expression patterns". Natural selection is surely acting across these time scales to produce some if not all of these evolutionary traits. Furthermore, it should not

be necessary to include additional common hosts to discuss these traits in these terms.

Line 54: "acquisition of fungal pathogenicity that allowed infection of vertebrates followed by evolution of host specificity". I am not convinced that we know the direction of evolution - and this has important implications here and other places in the manuscript. Since their divergence, *Bsal* could have evolved increased host specificity, *Bd* could have evolved increased generality, or both.

A) We agree that there are a multitude of possibilities that could result in contemporary traits. However, we have purposefully avoided describing directionality of these traits in this manuscript. Indeed, this sentence is intended to show that the differences of these two *Batrachochytrium* species offers a unique opportunity to shed light on the acquisition of fungal pathogenicity. We maintain that it does – and that additional pathogenic chytrids of Amphibians would also offer further opportunities. Given the recent discovery of *Bsal* – the opportunity of comparing even two is very recent – indeed this is the first– so feel it is appropriate to describe this as a motivation. As the only vertebrate infecting species in the Chytridiomycota, a parsimonious interpretation is that association with amphibians evolved first, followed by changed in host specificity. However, we have not implicitly described this or alternative evolutionary routes to pathogenicity in this paper.

Line 72: There is no mention of sequencing the *Tw* transcriptome in the Introduction, which seems odd because much of the discussion relies on the findings from that experiment.

A) The final paragraph of the introduction does mention the sequencing of *Tw*: "To more broadly characterize the host-pathogen interplay during infection, we compared the *in vivo* transcriptomes of *Bd* and *Bsal* in a susceptible model host species (*Tylotriton wenxianensis*, *Tw*) at late stage of infection against transcription *in vitro*".

Line 75: I am not convinced that the focus on "evolutionary adaptation" here and throughout is necessary. The authors have demonstrated important genomic changes, but not necessary adaptation per se (we don't know if these genomic patterns were driven by natural selection).

A) The divergence times between these species, and the abundance of genetic and phenotypic differences supports that many of these traits are the result of natural selection.

Line 81: Is it true that not only is the *Bsal* genome bigger than the *Bd* genome, but the *Bd* genome is also bigger than the other chytrids studied? It would be good to report in the main text the *Bd* genome size (is it 8,952?).

- A) Line 81 discusses the number of protein coding genes (which we say is reflected to the genome size). The *Bsal* genome is the largest of the four chytrids at 32.6 Mb. The *Bd* genome is 23 Mb. The number of transcripts that *Sp* has 8,952. We have now included the genome sizes for the *Batrachochytrium* species here.

Line 116: The terms “Amphibian/Caudate Destroying M36s” are unnecessarily overdramatic. The authors have not unambiguously determined the functional effect of these genes and gene families. “Associated” instead of “Destroying” would be much more appropriate.

- A) We feel it is appropriate to name this group according to their expected function, given the prior work on this family in other fungi and the experimental data we present.

Line 130: Example of place where the word “demonstrates” seems to overstep the data, “suggests” would be better.

- A) Changed as suggested.

Line 205-207: The findings are presented in a way that implies generality (e.g., that *Bsal* has immune dampening properties across caudates). The conclusions they can make from one host species are limited and this hypothesis will have to be tested in a broader range of host species.

- A) The final paragraph of the results details the host response from the two pathogens, which we found to be markedly different. While it is possible that more nuanced patterns would be identified from the inclusion of additional shared host species, we feel we have fairly described the patterns identified. To address the reviewer’s concerns, we have updated the text to allow some greater ambiguity as requested. We have not stated that we found *Bsal* has immune dampening properties, but our data suggests this is the case

Line 208: There should either be a new paragraph added at the end of the Results or Discussion section that addresses necessary caveats, or caveats should be integrated throughout the existing Results section in appropriate places (e.g., the need for ongoing studies in other host species, with other *Bd* isolates, possible biases with genome sequencing methods, etc.)

- A) We have now included an additional paragraph in the discussion, which details the diversity of host and pathogen and future work as suggested.

Line 211: The claim that the expansion of virulence factors “has expanded the host range...” is assigning causality and directionality to evolution that cannot be

supported by the data. Again, I would like to see a clear distinction between association and causality. For example, this could be modified to: "differential expansion of putative virulence factors by at least two chytrid fungi is associated with an expanded host range..."

- A) We have changed this sentence to say the expansion is associated, as suggested.

Line 214-220: This paragraph uses unnecessary dramatic and somewhat redundant language that oversells the findings. Examples include "impressively" "arsenal" and "armamentarium".

- A) *Bsal* has a significant increase in an array of resources available for a certain purpose (the definition of arsenal/armamentarium). We have dropped the more subjective "impressively" for "notable".

Line 234: This is the first mention of which strain of *Bd* is used in this study. This should be presented earlier and the choice of strain/acknowledgement of the diversity of *Bd* should be mentioned.

- A) *Bd* JEL423 is one of only two *Bd* strains that have been assembled into widely used reference genomes, and derive importantly from the Global Panzootic Lineage, which is considered to have been most devastating to amphibian populations. To address this comment, we have now included an additional paragraph in the discussion, which details the diversity of host and pathogen and possible future work.

Line 229-243: *Bsal* was sequenced with short reads and short inserts (<2.5kb) whereas *Bd* was sequenced with Sanger technology with 4kb, 10kb and 40 kb fosmids. I am concerned about whether there could be any artifacts that could arise from using two different methods for sequencing each genome. I would like to see a brief discussion about whether some genomic elements may be differentially represented in the assembly based on the sequencing approach.

- A) Based on our evaluation of the assemblies and gene sets for *Bd* and *Bsal*, we see that while the assembly of *Bsal* is more highly fragmented than the gene sets of both are highly complete. The more fragmented assembly prevents a deeper analysis of synteny between the two species. Other than connectivity, there could be some impact on repetitive element assembly, and we have commented in the discussion that an improved *Bsal* assembly could shed more light on repetitive element content.

Line 318, 321: "a transcriptome" would be more appropriate than "the transcriptome" given that this the sequencing is from a limited set of tissues

under particular conditions.

- A) The text was modified to note this is transcript sequence from one condition.

Line 358-378: I was quite confused about how zoospores were separated from sporangia. Line 358 states that all life stages were collected, but then line 375 talks about separate samples of spores vs sporangia. Please clarify methodological details and specific how pure the samples of each life stage are expected to be.

- A) Spores for both *Bd* as *Bsal* were collected by putting fresh water on a culture flask containing mature sporangia. Once the zoospores were released, the water containing the zoospores was collected. In order to reduce the percentage of mature cells, the water containing zoospores was passed over a sterile mesh filter with pore size 10 μm (Pluristrainer, PluriSelect). This information has been added to the Supplemental info lines 512-516.

Figure 1: Why is the figure arranged so that *Sp* is being compared to *Bsal*? It seems like it would be more phylogenetically informative and accurate to rotate the tree around that node so that *Sp* is compared to *Hp*.

- A) Node rotation between *Sp* to the other 3 chytrids could be done. However, a similarly small number of syntenic chains are identified between *Sp* and any of the other 3 chytrids – so that is the main finding with this distance, rather than a comparison between the pathogens and saprobes.

Figure 2-4: Figure 2, 3 and 4 are extremely crowded, confusing, and difficult to interpret. The most essential information should be conveyed in very parallel structure across figures. Below are just some of the questions I had, but I think the structure and content of these figures should be rethought from the ground up.

- A) We have made efforts to improve the clarity of each of the figures. To start with, we excluded the statistical analyses and placed these in a supplementary table. The size and spacing for Figure 2 has been improved to have a larger font size for the heat map. We have also excluded the heatmap entirely for the CBM18 figure, as this did not show a clear patterns, and was a previous recommendation.

Figure 2: What do the trees on top and to the side of the heat map represent? They should be either explained in the caption or removed. The labels to the right of the heat map are unreadable and could be simplified to just “*Bd*” or “*Bsal*.” The figures in c and d should be simplified or removed.

- A) The trees above and to the side of the heat maps are from hierarchical clustering – and therefore the top shows similarity between replicates, and the side shows similarity between genes. A mention of this is included in both figure 2 and figure 4, which also has heatmaps.

Figure 3: The representation of the gene structure seems unnecessary and makes the figure confusing. In general, there should be more of a parallel structure between the gene family heatmap figures to make all of them more interpretable (i.e., trees vs networks). The figures in c are hard to interpret.

- A) Reviewer 1 suggested that we remove the heatmap entirely based on the complex and inconsistent pattern of expression. We agreed that we could simply summarize this pattern instead of showing all values. Given that one of the key findings were the truncated Lectin-like CBM18 genes, which include a great reduction in CBM18 modules, we feel that the gene structure is informative and should remain. To address the comment on part c (which is now b), we have moved all the statistical analyses to a separate supplementary table.

Figure 4: The Bd CRN type letter plots seem unnecessary. Same comment about the bar graphs in figure c – they should be made more interpretable or removed to simplify the display.

- A) The motif plots show the conservation across the chytrid crinklers, which has not been previously shown. Similarly, the barplots show the first qPCRs for a range of crinklers, and at various life stages, showing significantly different expression values depending on life stage, and species. We think these analyses will be of interest to anyone interested in resolving their function.

The supplemental materials are long and complicated.

- A) We have categorized and renumbered the supplementary materials as noted in the response to previous comments.

REVIEWERS' COMMENTS:

Reviewer #1 (Remarks to the Author):

As I noted in my review of the first version of this manuscript, this article reports on the newly sequenced genome of *Batrachochytrium salamandrivorans* (Bsal) in comparison with the previously sequenced genome of *Batrachochytrium dendrobatidis* (Bd). The genomes of both pathogens were compared with those of two nonpathogenic chytrid species, *Homolaphlyctis polyrhiza* (Hp) and *Spizellomyces punctatus* (Sp). It adds additional RNA-seq data for expression patterns of both pathogens growing in liquid culture, in the skin of a susceptible host, and the skin response of the host salamander species (*Tylototriton wenxianensis*) following infection by both pathogens. The central findings are that Bd and Bsal share a number of gene families distinct from those shared with the two nonpathogenic chytrids that have a saprophytic lifestyle rather than infecting the skin. The unique gene families that are enriched in Bd and Bsal include a diverse array of M36 metalloprotease family members and Carbohydrate Binding Module 18 domain proteins. However, Bsal contains a larger genome than the other chytrids with expansion of several novel gene sets. When a vulnerable salamander, *T. wenxianensis*, was experimentally infected with each pathogen, the transcriptomic response of the host skin was greater against Bd than Bsal (more transcripts upregulated or down-regulated in comparison with skin from uninfected control animals), and many of the upregulated genes were for immune-related genes.

The findings are novel, and it is my view that publication of this information will provide a highly valuable resource for amphibian disease scientists in their efforts to understand the virulence characteristics of these unique and lethal fungal pathogens.

I thank the authors for addressing many of my previous concerns. However, there are a great many remaining problems with the current version. The most significant problems are with the integration of the supplemental information with the main text and with the way the data are presented and explained. I am keenly interested in the findings, yet I had to struggle with trying to understand this version of the manuscript nearly as much as the previous version. You have chosen to submit your work to a journal directed at a broad scientific audience rather than a journal for the specialist. If you want this work to be read and understood by the amphibian disease community, you need to do a better job of communicating what many of the figures and tables are intended to show. The supplemental figures and tables are not well connected to the main text. There are numerous examples in which there is a very general statement made with reference to multiple figures or tables listed in parentheses and no explanation of what each specific figure or table is intended to show and how it adds to the overall broad findings. The supplemental text is helpful, however perhaps the subsections should be numbered so that they can be linked to the main text when a greater explanation is needed. The supplemental methods are largely redundant with the information in the main text. In some cases, the methodological information presented in the main text is more complete. You should place the methods in one place or the other, but not both places. There also seems to be little effort to limit the length of the manuscript. It seems overlong and unwieldy. My recommendation would be to limit any text that is not absolutely necessary. It is often a

distraction rather than illuminating. I appreciate the care with which the genomic and transcriptomic studies were done and the need to demonstrate their validity, however it may not be necessary to list every step in the analysis. My specific concerns and suggestions are listed below with line number connections.

Specific Criticism or Suggestions:

Major points in Main Text:

1. Lines 26-39. The abstract was clearer when the results were presented in the past tense. I'm not sure why you switched tense unless it is a recommendation of the journal.
2. Lines 50-51. Change "markedly" to "apparently". Although the original study examining the Bsal susceptibility among anurans and caudates showed that only the caudates seemed to be susceptible, a limited number of anuran species were tested, and the infectious dose was quite low. Thus, some anuran species may yet be shown to be susceptible to infection and development of disease when exposed to Bsal. Therefore, it is the view of this reviewer that you should be more cautious about stressing that Bsal is a pathogen with a caudate-specific host range. In line 51, delete "being proscribed to" and replace it with "mostly infecting". "Proscribed" is not a useful term in this context.
3. Line 58. Replace "strategies and resulting lesions" with "outcomes". Host infection doesn't result in "strategies", but rather in different outcomes.
4. Line 67. Insert the word "is" after "(Sp)".
5. Line 78. Change "found" to present tense to match the present tense in the rest of the paragraph.
6. Lines 81-82. Delete "host-restricted" from line 81 and "the broad host range" from line 82. Here you are simply comparing the genome size. You don't need to emphasize species range here.
7. Line 83. Delete the reference to Table S1 here. It doesn't seem to have anything to do with genome size or complexity. It belongs in the methods section as you have indicated on line 260. Table S1 should probably be renumbered as the last supplemental table (S10) since it is not mentioned again until the methods section. Please also include the size of the genomes for Hp and Sp in this paragraph.
8. Lines 85-87. Please refer to Table S2 just after the word "chytrids" on line 85. This will lead the reader to the information you have just described in lines 83-85. Which specific figure shows the genes shared by Bd and Bsal of which 542 clusters are not found in the two nonpathogenic chytrids? Did you intend to refer to Fig. 1 here instead of Fig. S1. Are you referring to Fig. S1C here? Please be more explicit in leading the reader to the data supporting your statements.
9. Line 89. Which part of Fig. S1, Table S2, Table S3 or Table S4 makes the point about specific functions related to cell-wall modification and secreted effectors? Why refer to these tables and figures here unless you explain which figure panel makes the point? I see that Fig. S1D shows some genes (signalP4) suggesting they are secreted proteins. I found myself bouncing around between the main manuscript and supplements and still not seeing the information that you are trying to highlight. Please be more selective about where you refer to specific figures or tables. If you want to introduce the information here, then explain briefly what each table and figure and individual panel within a figure shows. The

figure legends should be amplified to spell this out more clearly. If you need to lead the reader to supplemental text, refer to the supplemental text by sub-section.

10. Line 90. Here, it might be useful to simply summarize what the phylogenetic tree (RAxML tree) and the synteny plot in Fig. 1 are intended to show, namely that Bd and Bsal are the most closely related to each other and both are also closely related to Hp. Finally, Fig 1 shows that Sp shares few genes with Bsal, and if the synteny plot was drawn to connect Sp to Bd or to Hp, it would share few genes with the other chytrids as well as you explained in the response to reviews.

11. In the legend for Fig. 1, please explain what each gray line connects. What do the smaller numbers along the genome length indicate?

12. Line 109. What is the significance of Tribe 3 and Tribe 1 genes and the specific transcripts shown in Fig. S3? In Fig. S3, are the differences in fold changes significant? If so, indicate with asterisks. Please insert a comma after "TGhL" in the legend. Please change "BD2432" in the legend to "BD24342" to match the figure itself.

13. Lines 115 and 117. Please insert hyphens between "species" and "specific".

14. Lines 121 and 123. I agree with reviewer 3 that "Amphibian Destroying M36s" and "Caudata Destroying M36s" are overly dramatic and may not be correct. I suggest instead "Batrachochytrium Fungal M36s (BFM) and "Batrachochytrium salamandrivorans M36s (BSM)" are more correct.

15. Lines 129-130. Please omit the references to Table S6 and Table S7. The data are shown in Fig. 2c-d, and it is a distraction to lead readers to Table S6 which shows the primer sets and Table S7 which shows the statistics on the fold changes. Those belong in the methods and the supplemental information. It may be OK to refer to Table S7 if you indicate in the main text that the statistical comparisons are shown in this supplemental table.

16. The legend for Table S7 is still somewhat unclear. To the bolded heading, please add: "The numerical values shown indicate fold change in comparison with freshly collected spores." I would also edit the rest of the legend as follows:

Significant changes between the target mRNAs are indicated with a superscript A for spores, B for Spores 2h + tissue, C for Sporangia, D for Sporangia + chitinase, and E for in vivo changes in salamander hosts. The use of these superscripts to show significance is still not entirely clear. For example, in the very first line of the "E: in vivo" column, why doesn't it include an "A" superscript to show that it differs significantly from expression in spores?

17. The legend for Fig. 2c indicates that significant differences will be shown with asterisks. However, there are no asterisks shown in the figure. For Fig 2d, are the differences shown significant?

18. Line 138. Change heading to read "Variation of cell-surface proteins in Bd and Bsal" since you are describing protein changes.

19. Line 140. Change "are" to "is".

20. Line 150. Please insert "Figure 3b" in place of just "Figure 3" to draw attention to panel b which illustrates the point about exposure to chitinases.

21. Line 158, add ", respectively" after "Hp".

22. Line 160. Break the sentence after the word "pathogens", and start the next sentence with "While".

23. Line 190. Delete "s" from "corroborates". Data is plural.

24. Line 198. If you introduce Table, S9, S10, and Fig. S8 here, you need to add some text

to explain what the figures show and how it adds to the narrative. Otherwise, don't refer to them here.

25. Line 218. Insert "putative" in front of "virulence factors".

26. Line 231. Insert "(JEL423)" to remind the readers that the information throughout refers to this BdGPL isolate.

27. Methods in main text. There are lots of acronyms tossed in. When first introduced, please define them. In the methods and legends, please be consistent in the use of these acronyms. For example, "RAxML" is used frequently, and often written as "RaxML". Choose one way to express it and be consistent.

28. Line 409, replace "aqua dest" with "distilled water".

29. Line 412, indicate which supplemental figure you mean, not "SX"

30. Figure legends. Please refer to "RAxML" trees as "Phylogenetic trees" or as "Phylogenetic (RAxML) trees".

31. Fig. 2c,d, and 3b legends. Either show significance with bars and asterisks (preferred) or add a final line "Significant differences in expression between each experimental group is shown in Table S7."

32. Legend for Fig 5, define "MA", "TMM", "FPKM".

Supplementary Information:

1. Line 265. The text refers to Figure 1D, but there is no Figure 1D. Please explain here the importance of "intergenic distances".

2. Line 319, Add "s" to deacetylase".

3. Line 328. There is no Figure 1B.

4. Lines 377-378. Why mention changes if they are not significant?

5. Line 384. "BSLG_07447" is designated "CBM1807447" in Fig. 3. Be consistent.

6. Lines 397-412 are entirely redundant with the methods in the main text.

7. Line 418 and elsewhere. Instead of using the acronym "cds" or "CDS", why not simply state coding sequence regions.

8. Line 419. Part of a sentence is missing.

9. Lines 425-436. This methods section is redundant with the methods in the main text.

10. Line 441. There is no Figure 1C.

11. Line 471. Did you intend to refer to Fig. 5C instead of Figure 2?

12. Line 474. Should this reference be to Fig. 5 instead of Fig. 2?

13. Line 484. Replace "clusters" with "clustered".

14. Line 494. There is no Figure 1A.

15. Lines 498-511. This is redundant with the main text, and in fact, the information in the main text is more complete.

16. Lines 515-535. This is redundant with the main text, and in fact, the information in the main text is more complete.

17. Lines 539-549. The information is redundant.

18. Legend for Fig. S2. Please define "MCL Tribes" in the legend.

19. Legend for Fig. S4. Please define "MEROPS"

20. Figure S5 is difficult to understand. Can it be explained more clearly? Is the main message that Bsal is predominantly a diploid species with some trisomy? There is no axis label in Fig S5b. What do the numbers indicate?

21. Figs S6 and S8 are still something of a mystery to me. Please add a few sentences to the end of the legends explaining what the figures are intended to show.

Reviewer #2 (Remarks to the Author):

Manuscript 102342_1 "Evolutionary innovations underpin niche-specific infection strategies across emerging pathogenic chytrid fungi" by Farrer et al.

In their revised manuscript, the authors have addressed most of the concerns I had about the first version. However, I'm a bit surprised that they didn't include at least a brief mentioning of previous genomic analyses of *Batrachochytrium* genomes as I suggested in point 5 in my comments. Even though a full comparative analysis might be beyond the scope of this manuscript, it would have been very helpful for readers to be pointed to additional information that is available on this species and to be given a brief summary on what is different in this study compared to previous work.

Reviewer #3 (Remarks to the Author):

I still find this to be a compelling study overall and the revisions address many of the previous reviewer comments. I have two (minor) new comments below. Addressing these two comments should be quick and will improve accuracy of the writing. I also flag some of the previous (again minor) comments that don't feel like they were fully addressed.

New comments:

Line 237: The authors state: Bd infection induces marked up-regulation of host genes involved in innate (i.e. inflammatory, antimicrobial peptides) and adaptive (i.e. immunoglobulin, MHC) immune responses, a feature of infection that has previously been noted in Bd.)" I am not convinced that the studies they reference (20, 21) show this conclusively. And there are plenty of other immunological and genomics studies that show no innate or adaptive immune response (or in fact a suppressed immune response) in frogs exposed to Bd. So this sentence should at least be toned down to indicate that Bd infection CAN induce IN SOME SPECIES... but that strong immune response is not a general feature of amphibian response to Bd.

Line 273: I appreciate the new caveats paragraph and I think all of that content should stay. However I would like to see two things made more explicit. At the end of (or after) the sentence "In this experiment, we have used a widely used isolate of Bd", the authors should also state "and we used a single host species". The following sentence also feels particularly awkwardly constructed. "However, other known lineages of Bd, as well as other shared hosts, may yield variation in induced immune responses alongside life-stage or infection-specific transcriptional responses". Perhaps more simply "However, other Bd lineages and

other hosts may respond differently, so generality will be improved by replicating this study with other Bd strains and other susceptible host species."

Previous comments that seem under-addressed.

I still found some of the uses of "adaptation" in the manuscript to be sloppy. I don't object to situating this work in a clearly evolutionary context, but the justification that the authors give in their rebuttal letter for why they do not take reviewer suggestions regarding the more stringent use of the term "adaptation" were unsatisfying to me. The authors say for example: "Natural selection is surely acting across these time scales to produce some if not all of these evolutionary traits." and "The divergence times between these species, and the abundance of genetic and phenotypic differences supports that many of these traits are the result of natural selection". In the field of evolutionary genomics, adaptation is **tested for**, not **assumed** based on evolutionary timescales or genetic or phenotypic differences. There are many molecular tests for adaptation that could be applied to the expanded gene families to look for signatures of natural selection at the genomic level. I understand if this is outside the scope of what the authors would like to do for this study, but it feels like an important point. Perhaps the most prominent place that this is problematic is the running head of "Adaptations that permit colonization of vertebrates radiate to allow divergent infection strategies in pathogenic chytrids".

Despite several reviewers commenting on the title seeming overblown, the authors did not want to change it. It strikes me that the authors are using "Evolutionary innovations" and "niche-specific" in a fairly general or colloquial sense whereas (as an ecologist and evolutionary biologist) those terms mean something very specific to me that requires more substantiation than presented in this study.

I still find it sensationalizing and unnecessary to name these "Amphibian Destroying M36s". The authors state in their rebuttal letter that "We feel it is appropriate to name this group according to their expected function", but no previous study has shown a specific way these M36s "destroy" amphibians.

I don't find the figures to be much more accessible than they were before.

Response to reviewer comments

Reviewer #1 (Remarks to the Author):

As I noted in my review of the first version of this manuscript, this article reports on the newly sequenced genome of *Batrachochytrium salamandrivorans* (Bsal) in comparison with the previously sequenced genome of *Batrachochytrium dendrobatidis* (Bd). The genomes of both pathogens were compared with those of two nonpathogenic chytrid species, *Homolaphlyctis polyrhiza* (Hp) and *Spizellomyces punctatus* (Sp). It adds additional RNA-seq data for expression patterns of both pathogens growing in liquid culture, in the skin of a susceptible host, and the skin response of the host salamander species (*Tylotriton wexianensis*) following infection by both pathogens. The central findings are that Bd and Bsal share a number of gene families distinct from those shared with the two nonpathogenic chytrids that have a saprophytic lifestyle rather than infecting the skin. The unique gene families that are enriched in Bd and Bsal include a diverse array of M36 metalloprotease family members and Carbohydrate Binding Module 18 domain proteins. However, Bsal contains a larger genome than the other chytrids with expansion of several novel gene sets. When a vulnerable salamander, *T. wexianensis*, was experimentally infected with each pathogen, the transcriptomic response of the host skin was greater against Bd than Bsal (more transcripts upregulated or down-regulated in comparison with skin from uninfected control animals), and many of the upregulated genes were for immune-related genes.

The findings are novel, and it is my view that publication of this information will provide a highly valuable resource for amphibian disease scientists in their efforts to understand the virulence characteristics of these unique and lethal fungal pathogens.

I thank the authors for addressing many of my previous concerns. However, there are a great many remaining problems with the current version. The most significant problems are with the integration of the supplemental information with the main text and with the way the data are presented and explained. I am keenly interested in the findings, yet I had to struggle with trying to understand this version of the manuscript nearly as much as the previous version. You have chosen to submit your work to a journal directed at a broad scientific audience rather than a journal for the specialist. If you want this work to be read and understood by the amphibian disease community, you need to do a better job of communicating what many of the figures and tables are intended to show. The supplemental figures and tables are not well connected to the main text. There are numerous examples in which there is a very general statement made with reference to multiple figures or tables listed in parentheses and no explanation of what each specific figure or table is intended to show and how it adds to the overall broad findings. The supplemental text is helpful, however perhaps the subsections should be numbered so that they can

be linked to the main text when a greater explanation is needed. The supplemental methods are largely redundant with the information in the main text. In some cases, the methodological information presented in the main text is more complete. You should place the methods in one place or the other, but not both places. There also seems to be little effort to limit the length of the manuscript. It seems overlong and unwieldy. My recommendation would be to limit any text that is not absolutely necessary. It is often a distraction rather than illuminating. I appreciate the care with which the genomic and transcriptomic studies were done and the need to demonstrate their validity, however it may not be necessary to list every step in the analysis. My specific concerns and suggestions are listed below with line number connections.

Response: In response to each of the specific points about confusion in accessing supplementary material, we have updated the call outs to refer more specifically to certain panels in figures and moved some call outs to the methods where more appropriate. Figure legends have also been updated to more clearly describe what is shown. Numbering of subsections of the supplementary information is not an option for this journal, however we have moved all methods to the main text. The overall length of the main text is already quite concise, however the noted duplications of methods text in the supplement have been removed.

Specific Criticism or Suggestions:

Major points in Main Text:

1. Lines 26-39. The abstract was clearer when the results were presented in the past tense. I'm not sure why you switched tense unless it is a recommendation of the journal.

Response: The journal style requires use of the present tense for the results in the Abstract.

2. Lines 50-51. Change "markedly" to "apparently". Although the original study examining the Bsal susceptibility among anurans and caudates showed that only the caudates seemed to be susceptible, a limited number of anuran species were tested, and the infectious dose was quite low. Thus, some anuran species may yet be shown to be susceptible to infection and development of disease when exposed to Bsal. Therefore, it is the view of this reviewer that you should be more cautious about stressing that Bsal is a pathogen with a caudate-specific host range. In line 51, delete "being proscribed to" and replace it with "mostly infecting". "Proscribed" is not a useful term in this context.

Response: Text replaced as suggested.

3. Line 58. Replace "strategies and resulting lesions" with "outcomes". Host infection doesn't result in "strategies", but rather in different outcomes.

Response: Text replaced as suggested.

4. Line 67. Insert the word “is” after “(Sp).

Response: Text added as suggested.

5. Line 78. Change “found” to present tense to match the present tense in the rest of the paragraph.

Response: Text replaced as suggested.

6. Lines 81-82. Delete “host-restricted” from line 81 and “the broad host range” from line 82. Here you are simply comparing the genome size. You don’t need to emphasize species range here.

Response: A simplistic expectation would be that a restricted pathogen would need a smaller genome compared to those that have broader host or niche ranges. As the opposite is true, that the restricted pathogen has the largest genome, we feel that this is worth noting as in this text.

7. Line 83. Delete the reference to Table S1 here. It doesn’t seem to have anything to do with genome size or complexity. It belongs in the methods section as you have indicated on line 260. Table S1 should probably be renumbered as the last supplemental table (S10) since it is not mentioned again until the methods section. Please also include the size of the genomes for *Hp* and *Sp* in this paragraph.

Response: The call out to Table S1 supports the clause at the start of this sentence: “By sequencing both *Bd* and *Bsal*”; we have moved the Table S1 callout to the end of this clause. The genome sizes of *Hp* and *Sp* were added as suggested.

8. Lines 85-87. Please refer to Table S2 just after the word “chytrids” on line 85. This will lead the reader to the information you have just described in lines 83-85. Which specific figure shows the genes shared by *Bd* and *Bsal* of which 542 clusters are not found in the two nonpathogenic chytrids? Did you intend to refer to Fig. 1 here instead of Fig. S1. Are you referring to Fig. S1C here? Please be more explicit in leading the reader to the data supporting your statements.

Response: The reference to Table S2 was moved earlier as suggested; the reference to Figure 1C was also added to the end of this sentence describing the 542 clusters.

9. Line 89. Which part of Fig. S1, Table S2, Table S3 or Table S4 makes the point about specific functions related to cell-wall modification and secreted effectors? Why refer to these tables and figures here unless you explain which figure panel makes the point? I see that Fig. S1D shows some genes (signalP4) suggesting they are secreted proteins. I found myself bouncing around between the main manuscript and supplements and still not seeing the information that you are trying to highlight. Please be more selective about where you refer to specific figures or tables. If you want to introduce the information here, then explain briefly what each table and figure and individual panel within a figure

shows. The figure legends should be amplified to spell this out more clearly. If you need to lead the reader to supplemental text, refer to the supplemental text by sub-section.

Response: We have revised the text and callouts at the end of this sentence to be more specific and refer to Figure 1C, Table S3, and Table S4. To the point raised by the reviewer, Figure 1C includes a summary of the orthogroups, Table S3 covers the categories of orthogroups, including the 542 not found in the chytrids, and Table S4 provides the details of the statistical tests on those orthogroups, showing the number of significant GO-terms and PFAM domains associated with each group, which provides evidence of genes are involved in cell-wall modification or secreted effectors for example.

10. Line 90. Here, it might be useful to simply summarize what the phylogenetic tree (RAxML tree) and the synteny plot in Fig. 1 are intended to show, namely that *Bd* and *Bsal* are the most closely related to each other and both are also closely related to *Hp*. Finally, Fig 1 shows that *Sp* shares few genes with *Bsal*, and if the synteny plot was drawn to connect *Sp* to *Bd* or to *Hp*, it would share few genes with the other chytrids as well as you explained in the response to reviews.

Response: We have added a description of these points to the end of this first paragraph of the results.

11. In the legend for Fig. 1, please explain what each gray line connects. What do the smaller numbers along the genome length indicate?

Response: The legend has been edited to address these points.

12. Line 109. What is the significance of Tribe 3 and Tribe 1 genes and the specific transcripts shown in Fig. S3? In Fig. S3, are the differences in fold changes significant? If so, indicate with asterisks. Please insert a comma after “TGhL” in the legend. Please change “BD2432” in the legend to “BD24342” to match the figure itself.

Response: The legend has been edited to address these points. These are genes selected arbitrarily to represent the tribe. A previous reviewer requested that we move significance notation to a table (previously significance was shown in the figure). Supplemental Table 7 provides significance between comparisons of conditions (shown under *Bd* extra and *Bsal* extra), and indeed these are significant between a number of conditions.

13. Lines 115 and 117. Please insert hyphens between “species” and “specific”.

Response: Text replaced as suggested.

14. Lines 121 and 123. I agree with reviewer 3 that “Amphibian Destroying M36s” and “Caudata Destroying M36s” are overly dramatic and may not be correct. I suggest instead “Batrachochytrium Fungal M36s (BFM)” and “Batrachochytrium salamandrivorans M36s (BSM)” are more correct.

Response: While we thank the reviewer for their suggestions to rename these groups, we maintain that our current names will provide the greatest interest among readers, in part due to the links provided by these names to host groups.

15. Lines 129-130. Please omit the references to Table S6 and Table S7. The data are shown in Fig. 2c-d, and it is a distraction to lead readers to Table S6 which shows the primer sets and Table S7 which shows the statistics on the fold changes. Those belong in the methods and the supplemental information. It may be OK to refer to Table S7 if you indicate in the main text that the statistical comparisons are shown in this supplemental table.

Response: The reference to Table S6 was removed from this section. The legends of Figures 2d, 3b, 4c and S3 were updated by including a reference to Table S7.

16. The legend for Table S7 is still somewhat unclear. To the bolded heading, please add: "The numerical values shown indicate fold change in comparison with freshly collected spores." I would also edit the rest of the legend as follows: Significant changes between the target mRNAs are indicated with a superscript A for spores, B for Spores 2h + tissue, C for Sporangia, D for Sporangia + chitinase, and E for in vivo changes in salamander hosts. The use of these superscripts to show significance is still not entirely clear. For example, in the very first line of the "E: in vivo" column, why doesn't it include an "A" superscript to show that it differs significantly from expression in spores?

Response: We agree that the table legend lacks clarity and adapted the legend according to the suggestions made by the referee as follows:

Supplementary Table 7: The numerical values shown indicate fold change \pm standard deviation in comparison with freshly collected spores. Superscript refers to a significant difference with the respective condition being "A" spores, "B" spores 2h + tissue, "C" sporangia, "D" sporangia + chitinase, and "E" *in vivo*. Gene targets included Amphibian Destroying M36s (ADM) and Caudata Destroying M36s (CDM), Carbohydrate binding-module 18 (CBM18), Crinklers (CRN), and Tribes with unknown function (extra). A Kruskal-Wallis analysis, followed by a Dunn-Bonferroni post hoc test was performed, indicating significant changes with $p < 0.05$. NA indicates that the condition was not tested.

In the first line, no superscript "A" was used for the *in vivo* condition because the change fold in mRNA expression in the *in vivo* condition was not statistically different from the fold change in the spores fraction. For example ADM_20379:

Pairwise Comparisons of VAR00001

Each node shows the sample average rank of VAR00001.

Sample1-Sample2	Test Statistic	Std. Error	Std. Test Statistic	Sig.	Adj.Sig.
3,00-2,00	5,667	4,082	1,388	,165	,991
3,00-1,00	11,833	4,082	2,899	,004	,022
3,00-4,00	-17,833	4,082	-4,368	,000	,000
2,00-1,00	6,167	4,082	1,511	,131	,785
2,00-4,00	-12,167	4,082	-2,980	,003	,017
1,00-4,00	-6,000	4,082	-1,470	,142	,850

1,00 = spore
 2,00 = spore + tissue
 3,00 = sporangia
 4,00 = in vivo

Each row tests the null hypothesis that the Sample 1 and Sample 2

parisons Test: Kruskal-Wallis Field(s): VAR00002 * VAR00001(Test 1)

The fact that this is not significant, although the upregulation is rather clear (27,36 vs 1,01), is probably the result of a large inter individual variation and the fact that the values are not normally distributed, resulting in less sensitive statistical tests. However, although this is not statistically significant, we are convinced of the biological importance of these differences. It is not a coincidence that we see an upregulation.

17. The legend for Fig. 2c indicates that significant differences will be shown with asterisks. However, there are no asterisks shown in the figure. For Fig 2d, are the differences shown significant?

Response: Asterisks had been incorrectly omitted in the revision, and now have been added back.

18. Line 138. Change heading to read “Variation of cell-surface proteins in Bd and Bsal” since you are describing protein changes.

Response: The heading was changed as suggested.

19. Line 140. Change “are” to “is”.

Response: The text was changed as suggested.

20. Line 150. Please insert “Figure 3b” in place of just “Figure 3” to draw attention to panel b which illustrates the point about exposure to chitinases.

Response: The text was changed as suggested.

21. Line 158, add “, respectively” after “Hp”.

Response: The text was changed as suggested.

22. Line 160. Break the sentence after the word “pathogens”, and start the next sentence with “While”.

Response: The text was changed as suggested.

23. Line 190. Delete “s” from “corroborates”. Data is plural.

Response: The text was changed as suggested.

24. Line 198. If you introduce Table, S9, S10, and Fig. S8 here, you need to add some text to explain what the figures show and how it adds to the narrative. Otherwise, don't refer to them here.

Response: This call out was modified to refer only to Table S9, which shows the Tw gene counts that are significantly up or down regulated by *Bsal* and *Bd* infection.

25. Line 218. Insert “putative” in front of “virulence factors”.

Response: The text was changed as suggested.

26. Line 231. Insert “(JEL423)” to remind the readers that the information throughout refers to this BdGPL isolate.

Response: The text was changed as suggested.

27. Methods in main text. There are lots of acronyms tossed in. When first introduced, please define them. In the methods and legends, please be consistent in the use of these acronyms. For example, “RAXML” is used frequently, and often written as “RaxML”. Choose one way to express it and be consistent.

Response: We have reviewed the methods for acronyms, however most capital letter terms refer to program names. The capitalization of RAXML has been corrected throughout the main and supplementary text.

28. Line 409, replace “aqua dest” with “distilled water”.

Response: Text replaced as suggested.

29. Line 412, indicate which supplemental figure you mean, not “SX”

Response: This table is now specified.

30. Figure legends. Please refer to “RAXML” trees as “Phylogenetic trees” or as “Phylogenetic (RAXML) trees”.

Response: The legends of Figure 1-4 were edited to refer to phylogenetic trees as suggested.

31. Fig. 2c,d, and 3b legends. Either show significance with bars and asterisks

(preferred) or add a final line “Significant differences in expression between each experimental group is shown in Table S7.”

Response: The legends have been updated as suggested.

32. Legend for Fig 5, define “MA”, “TMM”, “FPKM”.

Response: MA is not an acronym, and instead is an application of a Bland–Altman plot for visual representation of expression data which has been transformed onto the M (log ratios) and A (mean average) scale. TMM and FPKM have been given their full names as requested (The trimmed mean of M-values normalization method, and Fragments Per Kilobase of transcript per Million mapped reads).

Supplementary Information:

1. Line 265. The text refers to Figure 1D, but there is no Figure 1D. Please explain here the importance of “intergenic distances”.

Response: This has been corrected, as this referred to an additional panel in the previous version. Intergenic distances are described at the end of this section. In some plant pathogens, Crinklers are found with longer intergenic regions than other classes of genes, or what has become known as a “2X speed genome”. We have tested the same, as described in the main text.

2. Line 319, Add “s” to deacetylase”.

Response: The text was changed as suggested.

3. Line 328. There is no Figure 1B.

Response: Figure 1B has been changed to Figure 2.

4. Lines 377-378. Why mention changes if they are not significant?

Response: This sentence describes groups of changes in gene expression that are individually significant, but the groups maintains a similar pattern. It is arguably equally important to provide and not omit non-significant results from purposeful comparisons, providing that they are accurately described.

5. Line 384. “BSLG_07447” is designated “CBM1807447” in Fig. 3. Be consistent.

Response: Changed as suggested.

6. Lines 397-412 are entirely redundant with the methods in the main text.

Response: These paragraphs were deleted from the supplement.

7. Line 418 and elsewhere. Instead of using the acronym “cds” or “CDS”, why not simply state coding sequence regions.

Response: The text was changed as suggested throughout the supplementary text.

8. Line 419. Part of a sentence is missing.

Response: This section also appears redundant and was deleted from the supplement.

9. Lines 425-436. This methods section is redundant with the methods in the main text.

Response: These paragraphs were deleted from the supplement.

10. Line 441. There is no Figure 1C.

Response: This now refers to Fig. 2B.

11. Line 471. Did you intend to refer to Fig. 5C instead of Figure 2?

Response: Yes. Corrected.

12. Line 474. Should this reference be to Fig. 5 instead of Fig. 2?

Response: Yes. Corrected.

13. Line 484. Replace “clusters” with “clustered”.

Response: The text was changed as suggested.

14. Line 494. There is no Figure 1A.

Response: This was corrected refer to Figure 1.

15. Lines 498-511. This is redundant with the main text, and in fact, the information in the main text is more complete.

Response: These paragraphs were deleted from the supplement.

16. Lines 515-535. This is redundant with the main text, and in fact, the information in the main text is more complete.

Response: These paragraphs were deleted from the supplement.

17. Lines 539-549. The information is redundant.

Response: These paragraphs were deleted from the supplement.

18. Legend for Fig. S2. Please define “MCL Tribes” in the legend.

Response: MCL = Markov Cluster Algorithm. Added to legend.

19. Legend for Fig. S4. Please define “MEROPS”

Response: The MEROPS database is a resource for information on more than 4000 peptidases. MEROPS is not an acronym and just a name for this resource. We provide a link to the website already in both the main text and the supplementary text.

20. Figure S5 is difficult to understand. Can it be explained more clearly? Is the main message that Bsal is predominantly a diploid species with some trisomy? There is no axis label in Fig S5b. What do the numbers indicate?

Response: S5 compares measures of ploidy and repetitive content, as stated in the title. S5a indeed shows that *Bsal* is predominantly a diploid species with some possible trisomy, correct. Fig. S5b shows synteny between *Bsal* and *Bd*, colored according to allele-frequency. Given the contiguity of the *Bd* genome, it is clear that the trisomic regions of *Bsal* do not fall across an entire *Bd* chromosome, further suggesting *Bsal* has a diploid genome. We have added text to the manuscript clarifying these points.

21. Figs S6 and S8 are still something of a mystery to me. Please add a few sentences to the end of the legends explaining what the figures are intended to show.

Response: Fig. S6 shows intergenic regions of Crinklers. A sentence has been added stating that Crinklers in *Bd* have longer intergenic regions compared with the other 3 categories. Fig. S8 shows heatmaps of all the RNA-Seq datasets and replicates. This figure demonstrates replicability between datasets (correlation), and an overview across all the genes (overview); a new sentence has been added to the methods noting the high correlation between biological replicates.

Reviewer #2 (Remarks to the Author):

Manuscript 102342_1 "Evolutionary innovations underpin niche-specific infection strategies across emerging pathogenic chytrid fungi" by Farrer et al.

In their revised manuscript, the authors have addressed most of the concerns I had about the first version. However, I'm a bit surprised that they didn't include at least a brief mentioning of previous genomic analyses of *Batrachochytrium* genomes as I suggested in point 5 in my comments. Even though a full comparative analysis might be beyond the scope of this manuscript, it would have been very helpful for readers to be pointed to additional information that is available on this species and to be given a brief summary on what is different in this study compared to previous work.

Response: We thank the reviewer for these comments. We did include a few sentences regarding the population structure of *Bd*, and caveats on how additional lineages may yield different results. Furthermore, a number of gene families that we explore have been identified previously in *Bd* (i.e. M36s, CRNs), and citations to the prior work has been provided.

Reviewer #3 (Remarks to the Author):

I still find this to be a compelling study overall and the revisions address many of the previous reviewer comments. I have two (minor) new comments below. Addressing these two comments should be quick and will improve accuracy of the writing. I also flag some of the previous (again minor) comments that don't feel like they were fully addressed.

New comments:

Line 237: The authors state: Bd infection induces marked up-regulation of host genes involved in innate (i.e. inflammatory, antimicrobial peptides) and adaptive (i.e. immunoglobulin, MHC) immune responses, a feature of infection that has previously been noted in Bd.)" I am not convinced that the studies they reference (20, 21) show this conclusively. And there are plenty of other immunological and genomics studies that show no innate or adaptive immune response (or in fact a suppressed immune response) in frogs exposed to Bd. So this sentence should at least be toned down to indicate that Bd infection CAN induce IN SOME SPECIES... but that strong immune response is not a general feature of amphibian response to Bd.

Response: We have replaced those references with another reference (McMahon TA, Nature 2014) that found an immune response across multiple species. We have added the caveat that this is observed in some species.

Line 273: I appreciate the new caveats paragraph and I think all of that content should stay. However I would like to see two things made more explicit. At the end of (or after) the sentence "In this experiment, we have used a widely used isolate of Bd", the authors should also state "and we used a single host species". The following sentence also feels particularly awkwardly constructed. "However, other known lineages of Bd, as well as other shared hosts, may yield variation in induced immune responses alongside life-stage or infection-specific transcriptional responses". Perhaps more simply "However, other Bd lineages and other hosts may respond differently, so generality will be improved by replicating this study with other Bd strains and other susceptible host species."

Response: We have changed this sentence to be closer to the reviewer's suggestion. However, we feel that the specific differences that we anticipate are valuable to state, rather than invoking generality.

Previous comments that seem under-addressed.

I still found some of the uses of "adaptation" in the manuscript to be sloppy. I don't object to situating this work in a clearly evolutionary context, but the justification that the authors give in their rebuttal letter for why they do not take reviewer suggestions regarding the more stringent use of the term "adaptation" were unsatisfying to me. The authors say for example: "Natural selection is surely acting across these time scales to produce some if not all of these evolutionary traits." and "The divergence times between these species, and the abundance of genetic and phenotypic differences supports that many of these traits are the result of natural selection". In the field of evolutionary genomics, adaptation is *tested for*, not *assumed* based on evolutionary timescales or genetic or phenotypic differences. There are many molecular tests for adaptation that could be applied to the expanded gene families to look for signatures of natural selection at the genomic level. I understand if this is outside the scope of what the authors would like to do for this

study, but it feels like an important point. Perhaps the most prominent place that this is problematic is the running head of "Adaptations that permit colonization of vertebrates radiate to allow divergent infection strategies in pathogenic chytrids".
Response: We appreciate these comments, and understand that there are many further studies that can verify selection and adaptation, both within and between populations. The running header has been removed.

Despite several reviewers commenting on the title seeming overblown, the authors did not want to change it. It strikes me that the authors are using "Evolutionary innovations" and "niche-specific" in a fairly general or colloquial sense whereas (as an ecologist and evolutionary biologist) those terms mean something very specific to me that requires more substantiation than presented in this study.

Response: We have edited the phrases noted by the reviewer, changing the word evolutionary to genomic and removing the niche-specific phrase. To address the overarching concern that we have implied greater causality than strongly supported by our results, we also changed 'underpin' to 'linked to', which should mitigate the concerns of this reviewer.

I still find it sensationalizing and unnecessary to name these "Amphibian Destroying M36s". The authors state in their rebuttal letter that "We feel it is appropriate to name this group according to their expected function", but no previous study has shown a specific way these M36s "destroy" amphibians.

Response: We have now renamed these families Group 1 BatraM36 and Group 2 BsalM36, reflecting the phylogenetic conservation and clustering shown in Figure 2.

I don't find the figures to be much more accessible than they were before.

Response: Based on the reviewers' comments, we have made numerous changes to the figures. This includes reworking of some figures, moving statistical analyses into separate tables, removing heatmaps that were suggested to be unnecessary, and updating all main figure legends.